# Cytoplasmic protein misfolding titrates Hsp70 to activate nuclear Hsf1

Anna E Masser[1], Wenjing Kang[2], Joydeep Roy[1], Jayasankar Mohanakrishnan Kaimal[1], Jany Quintana-Cordero[1], Marc R Friedländer[2], Claes Andréasson[1]*

[1]Department of Molecular Biosciences, The Wenner-Gren Institute, Stockholm University, Stockholm, Sweden; [2]Department of Molecular Biosciences, The Wenner-Gren Institute, Science for Life Laboratory, Stockholm University, Stockholm, Sweden

**Abstract** Hsf1 is an ancient transcription factor that responds to protein folding stress by inducing the heat-shock response (HSR) that restore perturbed proteostasis. Hsp70 chaperones negatively regulate the activity of Hsf1 via stress-responsive mechanisms that are poorly understood. Here, we have reconstituted budding yeast Hsf1-Hsp70 activation complexes and find that surplus Hsp70 inhibits Hsf1 DNA-binding activity. Hsp70 binds Hsf1 via its canonical substrate binding domain and Hsp70 regulates Hsf1 DNA-binding activity. During heat shock, Hsp70 is out-titrated by misfolded proteins derived from ongoing translation in the cytosol. Pushing the boundaries of the regulatory system unveils a genetic hyperstress program that is triggered by proteostasis collapse and involves an enlarged Hsf1 regulon. The findings demonstrate how an apparently simple chaperone-titration mechanism produces diversified transcriptional output in response to distinct stress loads.

DOI: https://doi.org/10.7554/eLife.47791.001

*For correspondence:
claes.andreasson@su.se

**Competing interests:** The authors declare that no competing interests exist.

## Introduction

Heat shock factor 1 (Hsf1) is an ancient eukaryotic transcription factor that adjusts the cellular proteostasis system to changing stress loads by inducing the expression of a large set of genes (*Morimoto, 1998*; *Åkerfelt et al., 2010*). During acute heat shock, unfolded proteins accumulate in the cell resulting in the rapid and transient activation of Hsf1 that counteracts the harmful proteotoxic effects by inducing the expression of protective chaperones and aggregation factors (*Richter et al., 2010*). This core heat shock response (HSR) is a conserved transcriptional program that depends on the induced binding of Hsf1 as a trimer to promoters that harbour heat-shock elements (HSEs) (*Sorger and Nelson, 1989*; *Sorger and Pelham, 1987*; *Wiederrecht et al., 1987*). The target promoters also receive signals from other stress-induced pathways (*Verghese et al., 2012*). In budding yeast (*Saccharomyces cerevisiae*), the metabolic-stress transcription factors Msn2 and Msn4 (Msn2/4) associate with stress-response elements (STREs) present within many Hsf1 target promoters, leading to an expanded HSR that integrate stress-induced metabolic and proteostatic perturbations (*Gasch et al., 2000*; *Martínez-Pastor et al., 1996*; *Schmitt and McEntee, 1996*).

Regulation of Hsf1 activity to match the cellular load of unfolded proteins is central for maintaining proteostasis. In mammals, Hsf1 is kept latent by multiple layers of regulation, including cytosolic sequestration and monomerization, and stress triggers activation via nuclear targeting and induced DNA-binding (*Li et al., 2017*). Yeast Hsf1 is subject to more fundamental regulation and resides in the nucleus and is responsible for both basal and stress-induced gene expression (*Jakobsen and Pelham, 1988*; *Solís et al., 2016*; *Wiederrecht et al., 1988*). Hsf1 is considered to be regulated by a titratable chaperone repressor according to a model originally adapted from bacteria (*Craig and*

*Gross, 1991*). Accordingly, chaperones bind and repress Hsf1 and are titrated away by unfolded or misfolded proteins under stress conditions. The chaperones Hsp70 and Hsp90 have been implicated in the regulation of Hsf1 (*Abravaya et al., 1992*; *Bonner et al., 2000*; *Zou et al., 1998*). Recently evidence from the yeast model has been presented to support the notion that Hsp70 directly regulates Hsf1 as a titratable repressor (*Krakowiak et al., 2018*; *Zheng et al., 2016*). First, Hsp70 (Ssa1 and Ssa2) binds Hsf1 and this association is transiently decreased by heat shock. Second, overexpression of Hsp70 (Ssa2) together with its J-domain cochaperone Ydj1 attenuates Hsf1 activity. Finally, eliminating transcriptional induction of Hsp70 as part of the HSR results in prolonged Hsf1-activation following heat-shock. A peptide in the CE2 subdomain of Hsf1 has been implicated as an important regulatory site for the association of the Hsp70 repressor (*Høj and Jakobsen, 1994*; *Jakobsen and Pelham, 1991*; *Krakowiak et al., 2018*). Thus, Hsp70 functions as the titratable chaperone repressor of Hsf1.

A simple scenario that explains chaperone titration is based on the direct competition between Hsf1 and substrates for binding to the Hsp70 substrate-binding domain (SBD). Hsp70 transiently associates with hydrophobic peptides of unfolded and misfolded proteins via its SBD (*Rüdiger et al., 1997*). Initially, substrates weakly associate with the SBDβ of ATP-bound Hsp70 and induce ATP hydrolysis and then become trapped by the closing SBDα lid subdomain (*Mayer and Bukau, 2005*). Substrates in the cytosol are released by nucleotide exchange factors (NEFs) of the Hsp110 (Sse1 and Sse2) and armadillo (Fes1) classes that accelerate the exchange of ADP for ATP and thereby trigger opening of the SBD (*Bracher and Verghese, 2015*). Fes1 employs a specialized release domain (RD) to enable the efficient dissociation of persistent Hsp70 substrates such as misfolded proteins (*Gowda et al., 2018*). Notably, *fes1Δ* cells exhibit strong constitutive activation of Hsf1 suggesting that this NEF is involved in Hsf1 latency regulation at the level of substrate release from Hsp70 (*Gowda et al., 2013*; *Gowda et al., 2018*). Yet presently it is unclear whether Hsp70 handles Hsf1 as a canonical substrate and how NEFs influence the interactions. Moreover, the compartmentalized proteostasis system presents a spatial arrangement of significance for the chaperone titration model since both latent and transcriptionally active Hsf1 resides in the nucleus, while the bulk of its titratable negative regulator Hsp70, including the NEFs, are cytosolic and interact with newly translated proteins (*Morán Luengo et al., 2019*; *Sorger, 1991*).

In this study, we reconstitute the Hsf1-Hsp70 interaction and isolate activation complexes that bind HSEs and are negatively regulated by excess Hsp70. Hsp70 binds Hsf1 via its canonical SBD providing direct support for the model of direct competition between chaperone substrates and Hsf1. During heat shock, these activating substrates are derived from the misfolding of cytosolic translation products. Unleashing Hsf1 from Hsp70 control reveals a cryptic hyper-stress program with a widely broadened gene-target signature and much amplified transcriptional effects. Thus, our data provides mechanistic insight into how Hsf1 activity is regulated by Hsp70 to modify the HSR regulon.

## Results

### Reconstitution of large Hsf1-Hsp70 complexes with regulated HSE-binding activity

To investigate how Hsf1 is regulated by Hsp70, we reconstituted a complex of yeast Hsf1 and Hsp70 Ssa1 by coexpression in *E. coli* together with the J-domain protein Sis1. Following tandem-affinity purification with matrices specific for Ssa1 (6 × His SUMO; Ni²⁺-IDA) and Hsf1 (Strep Tag II; Strep-Tactin Sepharose) a complex with the apparent stoichiometry of Hsf1 and Ssa1 3:1 was isolated (*Figure 1A*, *Figure 1—figure supplement 1A*). Sis1 did not copurify with the complex but did as expected interact with Ssa1 in the first affinity purification step (*Figure 1—figure supplement 1B*) (*Horton et al., 2001*). The purified complex eluted as a single peak during size exclusion chromatography (SEC) with an estimated size close to 600–700 kDa (*Figure 1B*). Addition of increasing levels of ATP dissociated the complex quantitatively, consistent with the notion that Hsf1 is associated specifically with Hsp70-ADP and perhaps bound by its closed SBD (*Figure 1A*). To test the functionality of Hsf1 in the complex, we employed an electrophoretic mobility shift assay (EMSA) and assessed the capacity of the isolated complex to bind HSE (*Figure 1C*). The complex bound specifically to the 26 bp HSE-containing DNA fragment as evidenced by competition with unlabeled probes. Addition

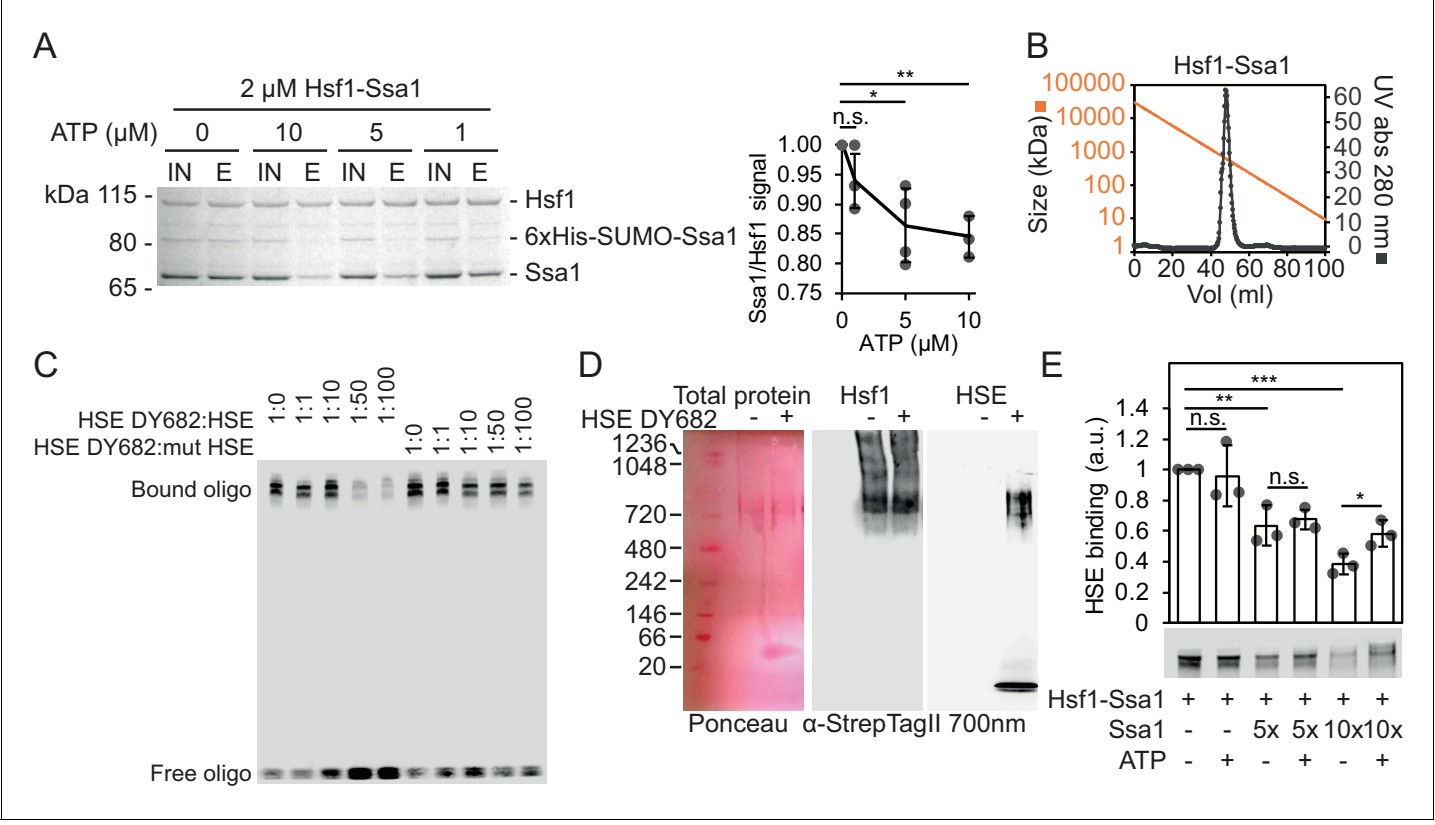

**Figure 1.** Reconstitution of large ATP-sensitive Hsf1-Hsp70 complexes with chaperone regulated HSE-binding activity. (A) Recombinant Hsf1-Ssa1 complexes (Input; IN) were immobilized onto Strep-Tactin Sepharose (StrepTag II in Hsf1) in the presence of 0, 1, 5 or 10 μM ATP and after washing, bound protein was eluted (E) with desthiobiotin. The Ssa1/Hsf1 densitometric signal ratios from silver staining were quantified. Error bars indicate standard deviation from at least three experiments. (B) Hsf1-Ssa1 complexes subjected to size exclusion chromatography. Molecular weight as a function of elution volume plotted in orange. (C) EMSA of fluorescently labeled HSE (HSE DY682) mixed with recombinant Hsf1-Ssa1complexes. The complexes were competed with unlabeled (HSE) or unlabeled and inactive mutant HSE (mut HSE). (D) Hsf1-Ssa1 complexes were incubated with and without HSE DY682 and were separated by native gel electrophoresis. HSE binding was assessed using 700 nm in-gel fluorescence and the Hsf1-Ssa1 complexes were visualized following membrane transfer (Ponceau S staining and α-StrepTag II). (E) Formation of Hsf1-Ssa1-HSE complexes (0.25 μM Hsf1-Ssa1) was assessed by EMSA in the presence of increasing concentrations of extra added Ssa1 (5x, +1.25 μM and 10x +2.5 μM) with or without 1 mM ATP. Error bars show standard deviation of at least three independent experiments.

DOI: https://doi.org/10.7554/eLife.47791.002

The following figure supplements are available for figure 1:

**Figure supplement 1.** The DNA-binding activity of recombinant Hsf1-Ssa1 complexes is inhibited by Hsp70.
DOI: https://doi.org/10.7554/eLife.47791.003
**Figure supplement 2.** Hsf1 forms supercomplexes upon supplementation with excess Hsp70.
DOI: https://doi.org/10.7554/eLife.47791.004

of the EMSA probe to native gels demonstrated that the intact Hsf1-Ssa1 complexes bind HSEs (*Figure 1D*). Consistent with this notion, addition of Ssa1-reactive serum resulted in that a fraction of the EMSA probe migrated as an ATP-sensitive supershifted smear (*Figure 1—figure supplement 1C*). In the same line, the Hsp70 Ssa2 was found to interact in vivo with the Hsf1-dependent promoter of *HSC82* by ChIP (*Figure 1—figure supplement 1D*). Interestingly, supplementation of the Hsf1-Ssa1 complexes with additional Ssa1 at low micromolar levels decreased HSE binding in a titratable manner as evidence by decreased signal of bound as well as of free oligonucleotides (*Figure 1E* and *Figure 1—figure supplement 1E–F*). With the exception of the condition in which the highest concentration of Hsp70 was applied, ATP supplementation did not significantly impact on the inhibitory effect that excess Hsp70 exerted on Hsf1 DNA binding (*Figure 1E* and *Figure 1—figure supplement 1F*). SEC analysis of Hsf1-Ssa1 complexes revealed that addition of Ssa1 resulted in the formation of Hsf1-containing supercomplexes that eluted earlier from the column than the

largest size marker (669 kDa) (*Figure 1—figure supplement 2*). Taken the characterization of the reconstituted Hsf1-Ssa1 complex together, Hsf1 resides in Hsp70 complexes that populate at least two conformations; activation complexes that are competent of binding HSEs and larger latency complexes that are unable to bind DNA. Excess Hsp70 pushes the equilibrium towards the latency complexes.

## Hsp70 engages Hsf1 via its substrate binding domain

To directly test the hypothesis that Hsp70 contacts Hsf1 via its SBD, we employed in vivo site-specific photo crosslinking (*Chen et al., 2007*). Briefly, the photoactivatable amino acid p-benzoyl-l-phenylalanine (pBPa) was incorporated by amber codon suppression at amino acid position 423 of the Ssa1 SBDβ (Ssa1$_{E423BPa}$). We have recently shown that Ssa1$_{E423BPa}$ readily crosslinks to bound substrates (*Gowda et al., 2018*). UV exposure of cells expressing Ssa1$_{E423BPa}$ resulted in widespread formation of crosslinked species that were detectable by western analysis as protein migrating larger than 70 kDa (*Figure 2A*). The overall crosslinking efficiency was not affected by a 3 min heat shock of the cells at 43°C prior to the UV irradiation. This shows that Hsp70 engages substrates under non-stressful as well as under stressful conditions. Western blotting demonstrated that Ssa1$_{E423BPa}$-bound Hsf1 via its SBD under non-stressful conditions and that the interaction between Ssa1 and Hsf1 exhibited a 77% decrease when cells were heat shocked (*Figure 2A–B*). As an alternative approach, we induced the misfolding of newly synthesized proteins by supplying the cultures with the proline ring analogue azetidine-2-carboxylic acid (AzC) (*Fowden et al., 1967*; *Trotter et al., 2002*). This condition that efficiently activates Hsf1, reduced the levels of crosslinking between Hsf1 and Ssa1$_{E423BPa}$ to undetectable levels (*Figure 2C–D*). Similarly, performing the assay in *fes1Δ* cells that constitutively activate Hsf1 and accumulate misfolded proteins resulted in undetectable crosslinking (*Figure 2E–F*). Thus, Hsp70 SBD interaction with Hsf1 decreases under Hsf1-inducing conditions explaining how misfolded proteins activate Hsf1 DNA-binding activity by titrating Hsp70 away from Hsf1-Hsp70 complexes.

## Nuclear Hsp70 maintains Hsf1 latent via canonical substrate binding

We set out to test the notion that nuclear Hsp70 hinders Hsf1 from binding HSEs. A prediction derived from this scenario is that increasing the release rates of bound Hsf1 from the nuclear pool of Hsp70 activates it. In the cell, release of Hsp70 substrates is accelerated by nucleotide exchange factors (NEFs) and in yeast they predominantly localize to the cytosol (*Verghese et al., 2012*). To increase Hsp70 Hsf1 release in the nucleus, we targeted the major cytosolic NEF Sse1 to the nucleus by fusing it to a strong nuclear localization signal (NLS) (*Figure 3A*). We determined the subcellular localization of Sse1 and Sse1-NLS. As predicted, Sse1-NLS localized in the nucleus, while Sse1 populated mainly the cytosol (*Figure 3B*).

Next, we directly tested Hsp70 substrate binding in cells expressing Sse1-NLS by employing the Ssa1$_{E423BPa}$ crosslinking assay. The overall crosslinking profile of Ssa1$_{E423BPa}$ remained unaltered in Sse1-NLS expressing cells suggesting that the bulk cytosolic Hsp70 substrate interactions were retained despite the increase nucleotide exchange in the nucleus. In contrast, Hsp70 crosslinking to Hsf1 was almost undetectable in Sse1-NLS expressing cells (*Figure 3C*). These findings are consistent with the scenario that Sse1-NLS accelerates specifically the release of substrates from Hsp70 in the nucleus, including Hsf1. To address this, we measured the activity of Hsf1 using a minimal promoter with HSEs that drive the expression of yeast Nanoluciferase (yNLuc) (*Masser et al., 2016*). Cells expressing Sse1-NLS displayed a threefold higher promoter activity compared to control cells expressing Sse1 carrying four point mutations (A280T, N281A, N572Y, E575A) that inactivate the NEF function (Sse1*-NLS), Sse1 with a nuclear export signal (Sse1-NES) or empty vector control (*Figure 3E*) (*Polier et al., 2008*).The effect was specific for Hsf1 since a derivative of the reporter with the HSEs exchanged to Msn2/4-responsive STRE elements did not display a change in activity when Sse1 was targeted to the nucleus (*Figure 3—figure supplement 1A*). Also, the Hsf1-dependent gene expression induced by Sse1-NLS was detectable on total protein level as increased levels of Hsp70, Hsp90 and Hsp104 (*Figure 3—figure supplement 1B*). Protein aggregate analysis showed that Sse1-NLS did not trigger protein aggregation suggesting that it does not hamper general protein folding (*Figure 3—figure supplement 1B*). A 30 min heat-shock at 37°C gave an even more striking effect on the Sse1-NLS-induced activation of Hsf1 as measured by *HSP104* mRNA levels

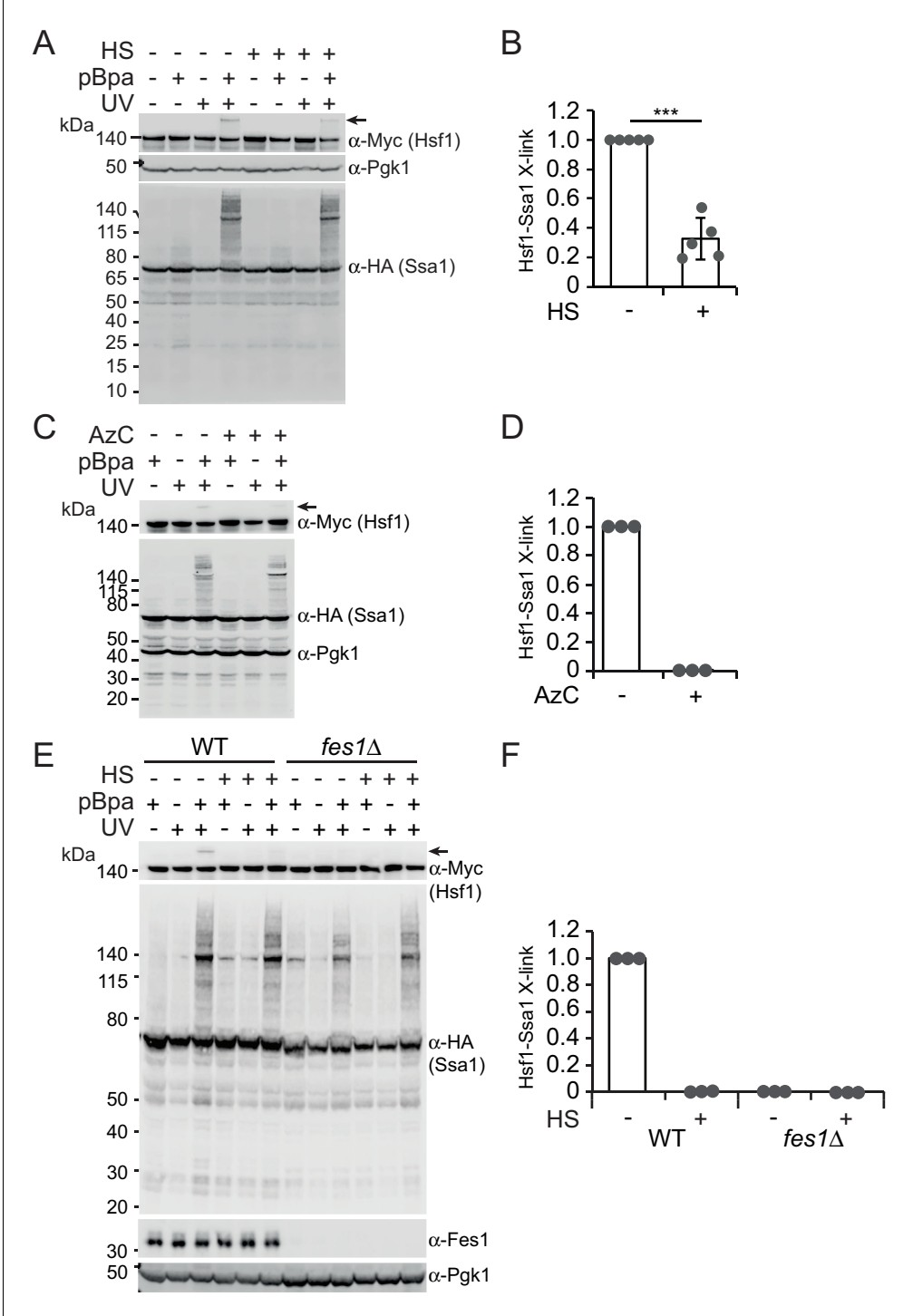

**Figure 2.** Heat shock negatively regulates Hsp70 binding of Hsf1 via its SBD. (A) Photocrosslinking (UV) of yeast cells grown at 30°C or subjected to a 3 min heat shock (HS) at 43°C that express Ssa1-HA with photoreactive Bpa (pBPa) incorporated at amino acid position 423 of the Hsp70 SBDβ in cells. Proteins and crosslinking-products were visualized by western analysis using α-HA and α-Myc antibodies with α-Pgk1 as a loading control. Representative data from five independent experiments are shown and the relative levels of Hsf1-Ssa1 crosslinking products marked with an arrow were quantified in (B) with error bars showing standard deviation. (C) and (D) The experiment in A was performed with a 2 hr AzC treatment at 25°C to induce the misfolding of newly synthesized proteins. (E) and (F) The experiment in A was performed using *fes1Δ* cells.
DOI: https://doi.org/10.7554/eLife.47791.005

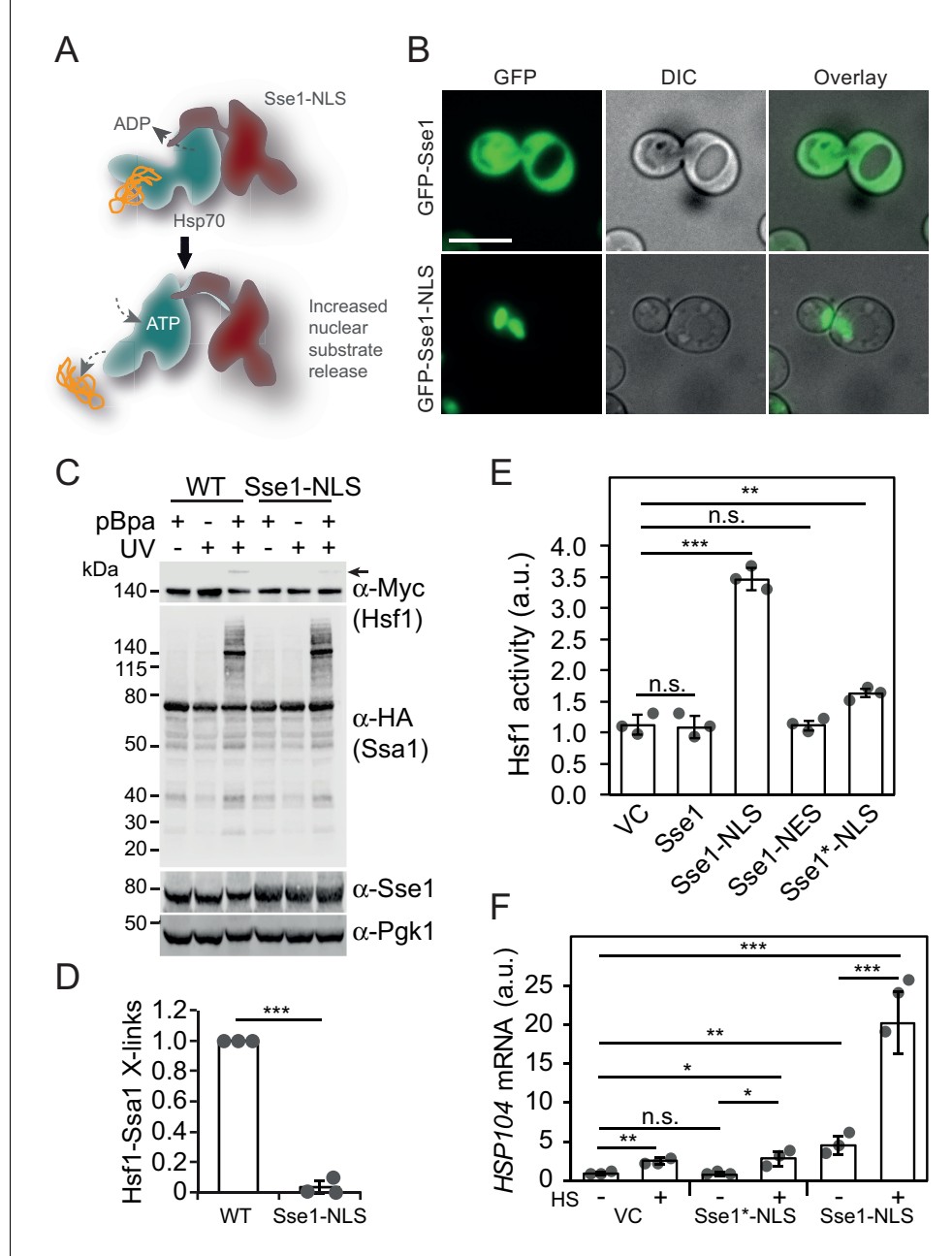

**Figure 3.** Accelerating Hsp70 substrate release in the nucleus activates Hsf1 and potentiates the heat-shock response. (**A**) Schematic representation of Sse1-NLS binding to nuclear Hsp70. Binding accelerates Hsp70 nucleotide exchange and triggers substrate release. (**B**) Micrographs of the subcellular localization of control GFP-Sse1 and GFP-Sse1-NLS. The scale bar corresponds to 5 μm. (**C**) Photocrosslinking (UV) of yeast cells grown at 30° C or subjected to a 3 min heat shock (HS) at 43°C that express Ssa1-HA with photoreactive Bpa (pBPa) incorporated at amino acid position 423 of the Hsp70 SBDβ in cells. Proteins and crosslinking-products were visualized by western analysis using α-HA and α-Myc antibodies with α-Pgk1 as a loading control. (**D**) The relative levels of Hsf1-Ssa1 crosslinking products (marked with an arrow in C) were quantified. (**E**) Hsf1 activity at 30°C in cells carrying vector control (VC) or plasmid derivatives that express Sse1, Sse1-NLS, Sse1-NES and Sse1*-NLS. Hsf1-activity was determined using a bioluminescent reporter construct. (**F**) Analysis of *HSP104* mRNA levels normalized to *TAF10* mRNA levels at 25°C and after a 30 min heat shock (HS) at 37°C in cells carrying the same plasmids as in E. All experiments were repeated three times with error bars showing standard deviation.
DOI: https://doi.org/10.7554/eLife.47791.006

The following figure supplement is available for figure 3:

*Figure 3 continued on next page*

*Figure 3 continued*

**Figure supplement 1.** Expression of Sse1-NLS does not affect Msn2/4 activity but strongly induces the expression of Hsp70, Hsp90 and Hsp104.

DOI: https://doi.org/10.7554/eLife.47791.007

(*Figure 3F*). While Sse1-NLS expressing cells under non-stressful conditions exhibited 5.5-fold higher transcript levels than cells with vector control or Sse1\*-NLS, a transient heat shock revealed hyperactivation of Hsf1 with transcript levels 24.4-fold above the basal control levels. In contrast, vector control and Sse1\*-NLS cells displayed a modest increase of *HSP104* transcript levels of 3.0- and 4.5-fold, respectively. The observed hyperactivation of Hsf1 and the induced loss of interaction with Hsp70 by Sse1-NLS demonstrate that specifically nuclear Hsp70 binding is required for Hsf1 latency control under both normal and stress conditions.

## Heat-shock activates Hsf1 by the misfolding of newly synthesized proteins

During heat shock, protein misfolding is considered the trigger for activation of Hsf1 via Hsp70 chaperone titration. We set out to investigate the role that newly translated proteins play in heat-shock-induced activation of Hsf1. First, we induced misfolding of newly synthesized proteins by transiently feeding cells with the proline ring analogue AzC. Consistent with earlier studies, we found that misfolding of newly translated proteins by AzC-induced Hsf1 activation within the first 30 min (*Figure 4—figure supplement 1A*) (*Trotter et al., 2002*). Pleotropic stress-activation of Msn2/4 was observed first 60 min after the AzC addition. In contrast, 30 min heat-shock at 37°C rapidly activated both Hsf1 and Msn2/4-dependent transcription with a peak in intensity after 10 min (*Figure 4—figure supplement 1B*). Combining the treatment of AzC and heat-shock revealed strong positive synergistic effects on Hsf1 activation, as determined by qPCR analysis of *SSA4* and *HSP104* transcripts (*Figure 4—figure supplement 1C*). Thus, increasing the pool of newly translated proteins that are unable to fold properly due to incorporation of AzC results in more efficient heat-shock-induced titration of Hsp70.

We directly tested the impact that decreasing of the pool of newly synthesized proteins has on heat-shock-induced activation of Hsf1. Cycloheximide (CHX) was added to arrest translation right before a 15 min heat shock. Newly translated proteins constitute the main source of aggregating species in growing yeast cells and using the Hsp104-GFP marker we could confirm that CHX efficiently blocked the accumulation of heat-induced protein aggregates (*Figure 4A–B*) (*Zhou et al., 2014*). Importantly, under identical experimental conditions we observed a fourfold reduction of *SSA4* transcript levels when arresting translation (*Figure 4C*). As an alternative experimental approach to reduce ongoing translation, we acutely starved *leu2* auxotrophic cells for leucine (*Figure 4D*). After 30 min of leucine starvation, translation rates had dropped significantly as evidenced by much decreased levels of the short-lived protein yNlucPEST expressed from the constitutive *TDH3* promoter (*Figure 4E*, *Figure 4—figure supplement 2A–B*). Under these conditions, heat shock at 37°C did not support any detectable Hsp104-GFP foci formation (*Figure 4F*) and importantly, resulted in fivefold lower levels of *SSA4* transcripts compared to the non-starved control (*Figure 4G*). In summary, inducing misfolding of newly synthesized proteins is sufficient to activate Hsf1 and ongoing protein synthesis is required for its heat-shock-induced activation. Our data suggest that the aggregation-prone character of newly synthesized proteins makes these species highly proficient in titrating Hsp70 away from nuclear Hsf1 latency complexes.

## Persistent Hsp70 substrates activate Hsf1

Aggregation-prone misfolded proteins display many hydrophobic Hsp70-binding sites and specifically depend on the armadillo-type NEF Fes1 for their efficient release from the Hsp70 SBD (*Gowda et al., 2016*; *Gowda et al., 2013*; *Gowda et al., 2018*). To more generally test if substrates with persistent Hsp70 binding are responsible for activating Hsf1, we inactivated Fes1 (*Figure 5A*). In our previous characterization of the *fes1Δ* transcriptional profile, we had found a clear Hsf1 activation signature in cells grown under otherwise non-stressful conditions (*Gowda et al., 2016*). In direct comparison with other inactivating mutations in the chaperone network, *fes1Δ* cells exhibited the

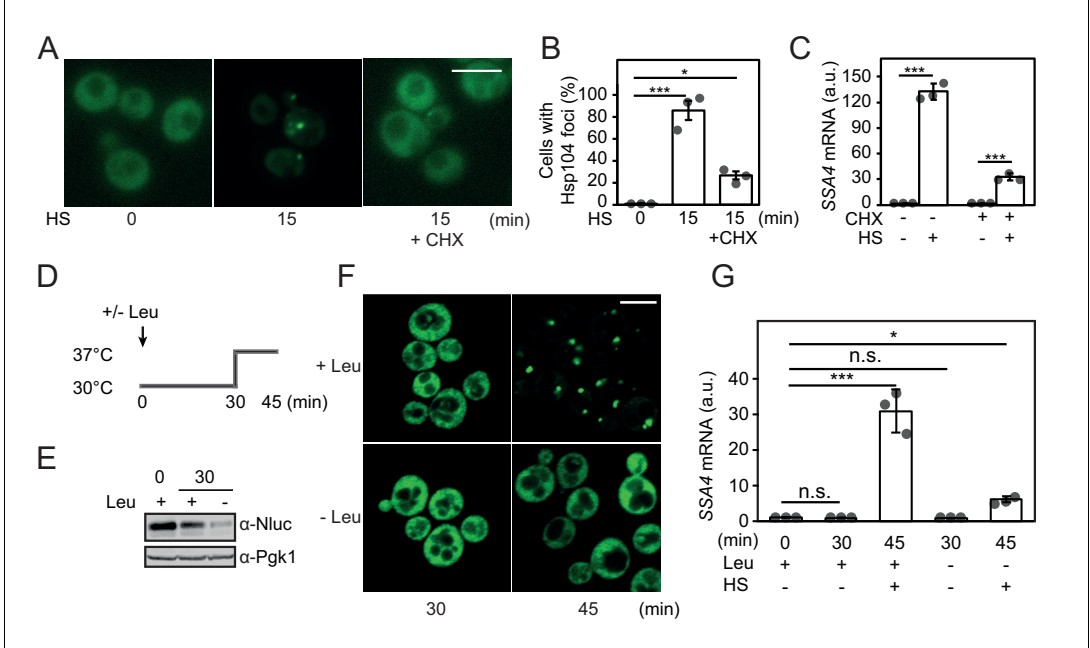

**Figure 4.** Heat shock activates Hsf1 by the misfolding of newly synthesized proteins. (**A**) Micrographs of Hsp104-GFP in cells grown at 25°C and heat shocked (HS) for 15 min at 37°C either with or without the addition of CHX. The white scale bar is 5 μm. (**B**) Quantification of the fraction of cells with Hsp104-GFP foci in A. At least 260 cells were counted in three biological replicates for each condition. Error bars denote standard error. (**C**) Quantification of *SSA4* mRNA levels in cells treated as in A. (**D**) Outline of the experimental setup for acute leucine starvation of *leu2Δ0* cells in E-G. (**E**) Steady state levels of the rapidly turned-over protein yNlucPEST expressed from the strong *TDH3* promoter in *leu2Δ0* cells grown in complete synthetic media or acutely starved for leucine for 30 min. (**F**) Micrographs of Hsp104-GFP in *leu2Δ0* cells grown under leucine-rich (+Leu) and acute leucine starvation (-Leu) conditions before (30 min) and after (45 min) a 15 min heat shock. The white scale bar is 5 μm. (**G**) Quantification of *SSA4* mRNA levels in cells grown at 30°C and heat-shocked (HS) for 15 min at 37°C with or without 30 min of leucine starvation. All qPCRs were normalized to *TAF10* mRNA levels and all bioluminescence measurements to $OD_{600}$. Error bars show standard deviation unless otherwise stated.

DOI: https://doi.org/10.7554/eLife.47791.008

The following figure supplements are available for figure 4:

**Figure supplement 1.** Heat shock activates Hsf1 by the misfolding of newly synthesized proteins.
DOI: https://doi.org/10.7554/eLife.47791.009
**Figure supplement 2.** Half-life of NlucPEST.
DOI: https://doi.org/10.7554/eLife.47791.010

highest induction of the HSR under non-stressful conditions (*Figure 5B*). The activation was specific for Hsf1 and Msn2/Msn4 activity remained at basal levels (*Figure 5C*). The role of Fes1 in maintaining Hsf1 latent was dependent on its NEF function, since abolishing the interaction with Hsp70 using two well-characterized amino acid substitutions (*fes1-1*; A79R, R195A) resulted in Hsf1 activation comparable to *fes1Δ* (*Figure 5D*) (*Shomura et al., 2005*). Thus, Fes1-mediated release of persistent substrates from Hsp70 is a requisite for Hsf1 latency regulation.

## Unleashing Hsf1 from Hsp70 control exposes a hyper-stress transcriptional program

To learn more about how persistent Hsp70 substrates influence Hsf1 activation, we heat-shocked *fes1Δ* cells 30 min at 37°C and performed RNA sequence (RNA-seq) analysis (*Figure 5—source data 1*). Principal component analysis showed that the experiment displayed excellent reproducibility and sample quality (*Figure 5E*). Following heat shock, most (86%) of the differentially expressed genes in wildtype cells were also differentially expressed in *fes1Δ* cells. Yet in *fes1Δ* cells an additional 860 genes were differentially expressed indicating an overall similar but stronger and broader effect in the mutant (*Figure 5F*). For the well-known Hsf1-target genes *HSP42, HSP82, HSP104, SSE1* and *SSA4* that encode constituents of the stress-inducible chaperone machinery the amplification of the heat shock response in *fes1Δ* cells was particularly accentuated and exhibited hyperinduction

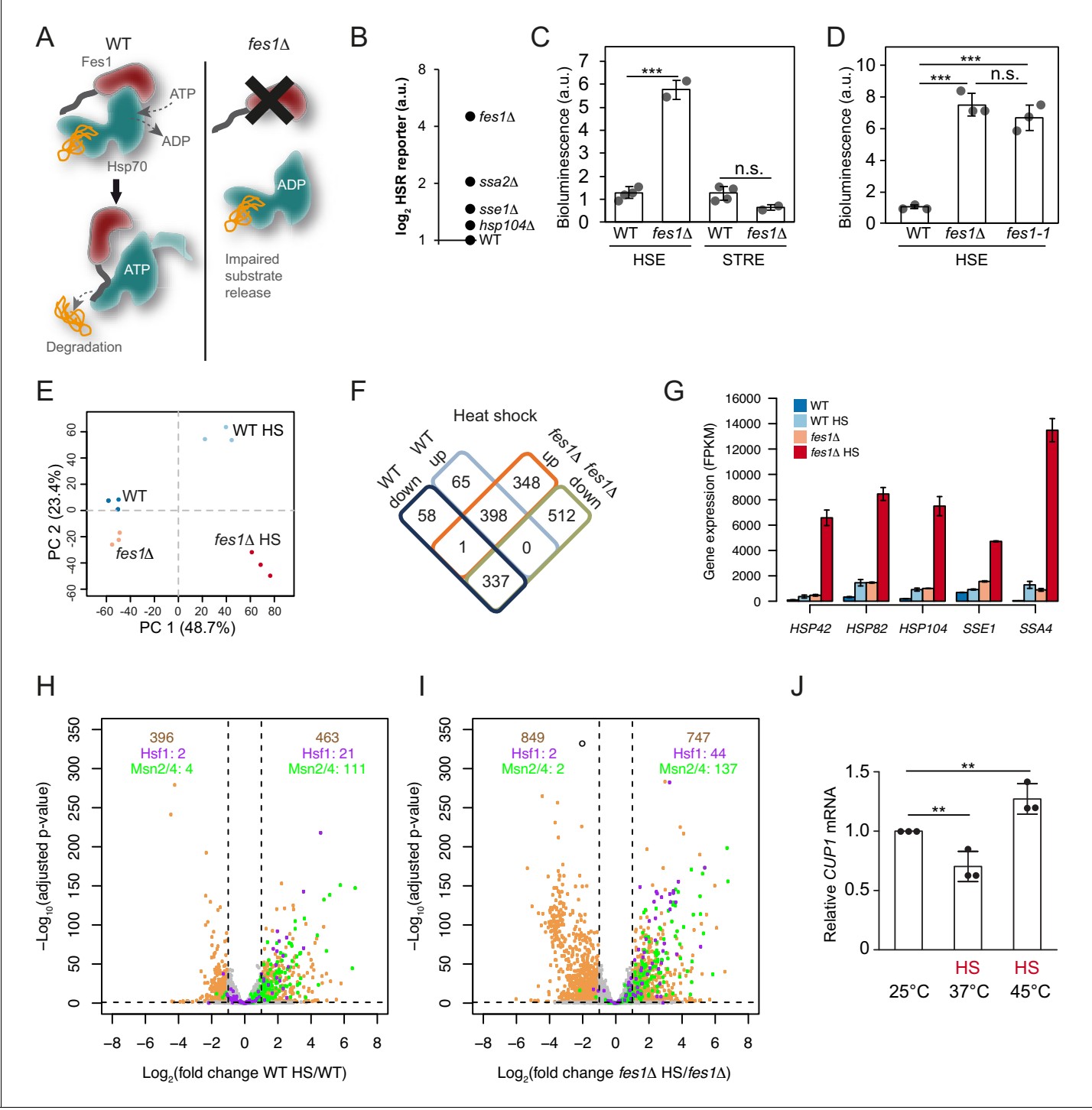

**Figure 5.** Limiting the available pool of Hsp70 by impairing cytosolic substrate release unleashes a Hsf1 hyper-stress program. (**A**) Schematic representation of Fes1-accelerated release of persistent substrates from Hsp70. In *fes1Δ* cells persistent substrates remain associated with Hsp70. (**B**) Hsf1 activity was determined in cells with chaperone mutations (*hsp104Δ*, *sse1Δ*, *ssa2Δ* and *fes1Δ*) grown at 25°C a using HSE bioluminescent reporter. (**C**) Hsf1 and Msn2/4 activities in WT and *fes1Δ* cells grown at 25°C were measured using HSE and STRE bioluminescent reporters. (**D**) Hsf1 activity was determined in WT, *fes1Δ* and *fes1-1* cells as in B. (**E**) Principal component analysis of transcriptome RNA-seq data of WT and *fes1Δ* cells grown at 25°C or subjected to heat shock (HS) for 30 min at 37°C. (**F**) Venn diagram of differentially expressed (DE) genes following heat shock in WT and *fes1Δ* cells. (**G**) Transcript levels (FPKM) of the Hsf1 target genes *HSP42*, *HSP82*, *HSP104*, *SSE1* and *SSA4*. (**H**) Volcano plot showing gene expression changes following heat shock of WT cells. The Hsf1 and Msn2/4 targets are colored in purple and green, respectively. Genes with adjusted p value < 0.05 and absolute log2 fold change >1 are considered significant. These thresholds are indicated by the dotted black lines. The number of genes that are up-

*Figure 5 continued on next page*

*Figure 5 continued*

and down-regulated are written in tan color at the top right and top left of the plot, respectively. Among these DE genes, the number of Hsf1 and Msn2/4 targets are showed below in purple and green color. (I) Volcano plot as in H but for *fes1Δ* cells. (J) Relative *CUP1* mRNA levels in WT cells grown at 25°C or subjected to heat shock (HS) at 37°C or 45°C for 30 min. Experiments were performed in triplicates with error bars showing standard deviation.

DOI: https://doi.org/10.7554/eLife.47791.011

The following source data and figure supplements are available for figure 5:

**Source data 1.** FKPM values from RNA seq analysis of heat-shocked WT and *fes1Δ* cells.

DOI: https://doi.org/10.7554/eLife.47791.016

**Figure supplement 1.** Induction of Hsf1 and Msn2/4 target genes under heat shock and hyper-stress conditions.

DOI: https://doi.org/10.7554/eLife.47791.012

**Figure supplement 2.** Msn2/4 is dispensable for hyperinduction of the HSR and expression of the *CUP1* gene.

DOI: https://doi.org/10.7554/eLife.47791.013

**Figure supplement 3.** Density plot obtained from DESeq2 analysis showing gene expression changes following heat shock in (A) WT and (B) *fes1Δ* cells.

DOI: https://doi.org/10.7554/eLife.47791.014

**Figure supplement 4.** GO analysis of differentially expressed genes in heat shocked cells.

DOI: https://doi.org/10.7554/eLife.47791.015

characteristics (*Figure 5G* and *Figure 5—figure supplement 1A–B*). For example, the highly Hsf1-responsive Hsp70 gene *SSA4* exhibited 30-fold induction in WT cells upon heat shock but was despite elevated basal expression levels induced 310-fold in *fes1Δ* cells. Similarly, aggregation factor *HSP42*, the disaggregase *HSP104*, the Hsp110 *SSE1* and the Hsp90 chaperone *HSP82* all exhibited an up to 5-fold heat-shock induction in WT cells. Yet, in heat shocked *fes1Δ* cells their transcriptional levels increased up to 70-fold. The hyperinduction of *SSA4* and *HSP104* in *fes1Δ* cells were not dependent on Msn2 and Msn4, demonstrating that the effect was the result of specifically Hsf1 activity (*Figure 5—figure supplement 2*). Inspection of the expression of all genes in WT and *fes1Δ* cells before and after the heat shock, revealed a highly accentuated transcriptional response in *fes1Δ* cells involving amplified induction and inclusion of more Hsf1 and Msn2/4 target genes (*Figure 5H–I* and *Figure 5—figure supplement 3A–B*). For Hsf1 targets genes, 44 were significantly induced in heat shocked *fes1Δ* cells while only 21 were induced in WT cells (*Pincus et al., 2018*). The corresponding numbers for Msn2/4 target genes were 137 and 111, respectively (*Solís et al., 2016*). Looking at the total number of heat-shock-induced genes, 463 were found to be induced in WT cells while 747 were induced in *fes1Δ* cells. The effect was even more pronounced for repressed genes with 396 identified genes in WT cells and 849 in *fes1Δ*. GO analysis revealed that *fes1Δ*-specific induction by heat shock included genes related to cellular detoxification, response to toxic substances, oxidant detoxification and carbohydrate metabolism (*Figure 5—figure supplement 4A–C*). The downregulated genes were related to the translation machinery, including tRNA and ribosome biogenesis as well as nucleotide metabolism. WT control cells displayed characteristic heat shock induced changes in metabolic transcription and upregulation of genes involved in the proteostasis network and DNA repair (*Gasch et al., 2000*; *Hahn and Thiele, 2004*). Repression of transcripts that encode ribosomal proteins is a hallmark of the yeast heat shock response and depends of a set of transcriptional regulators that functions parallel to Hsf1 (*Causton et al., 2001*; *Gasch et al., 2000*; *Rudra et al., 2005*; *Wade et al., 2004*). Overall, the behavior indicates that *fes1Δ* cells mount a hyper-stress program in response to heat-shock as a result of more induced damage by the treatment or as a result of sensitized regulatory circuits, for example severe Hsp70 out-titration.

We asked if also wildtype cells could access the hyper-stress program by analyzing the transcript levels of the diagnostic *CUP1* gene under more extreme heat-shock conditions. In wild-type cells, the RNA-seq analysis showed that *CUP1* responds to heat shock at 37°C by transcriptional downregulation, yet in *fes1Δ* cells it is induced by heat shock as part of the hyper-stress program (*Figure 5—figure supplement 2*). *CUP1* transcript analysis in wild-type cells by qPCR replicated the downregulation of transcription at 37°C and revealed that more extreme heat-shock conditions (45°C for 30 min) allowed also wild-type cells to induce the gene (*Figure 5J*). The *CUP1* promoter contains a minimal low-affinity HSE, suggesting that activation of the hyper-stress program involves increased levels of active Hsf1 due to severe out-titration of Hsp70 (*Sewell et al., 1995*). Thus, the accumulation

of persistent Hsp70 substrates changes Hsf1-dependent as well as other stress regulation to display hyper-induction/repression characteristics with greatly amplified transcriptional effects both when considering gene targets and induction amplitudes.

## Heat shock titrates the soluble pool of Hsp70

We asked if the Hsf1 activation and also the hyper-stress activation was the result of reducing the soluble Hsp70 pool by sequestration by aggregation-prone proteins. We performed protein-aggregate analysis by lysing cells using a high-pressure homogenizer followed by differential centrifugation of the lysates supplemented with nonionic detergent. Heat shocking WT cells at 37°C resulted in a somewhat changed pattern of proteins in the aggregate fraction compared to the unstressed 25°C condition (*Figure 6A–B*). Importantly, a prominent band migrating at 70 kDa was enriched in the aggregate fraction and was identified as Hsp70 using anti-Ssa1 antibodies (*Figure 6C*). Analysis of soluble and aggregate fractions showed that soluble Ssa1 was transferred to the aggregate

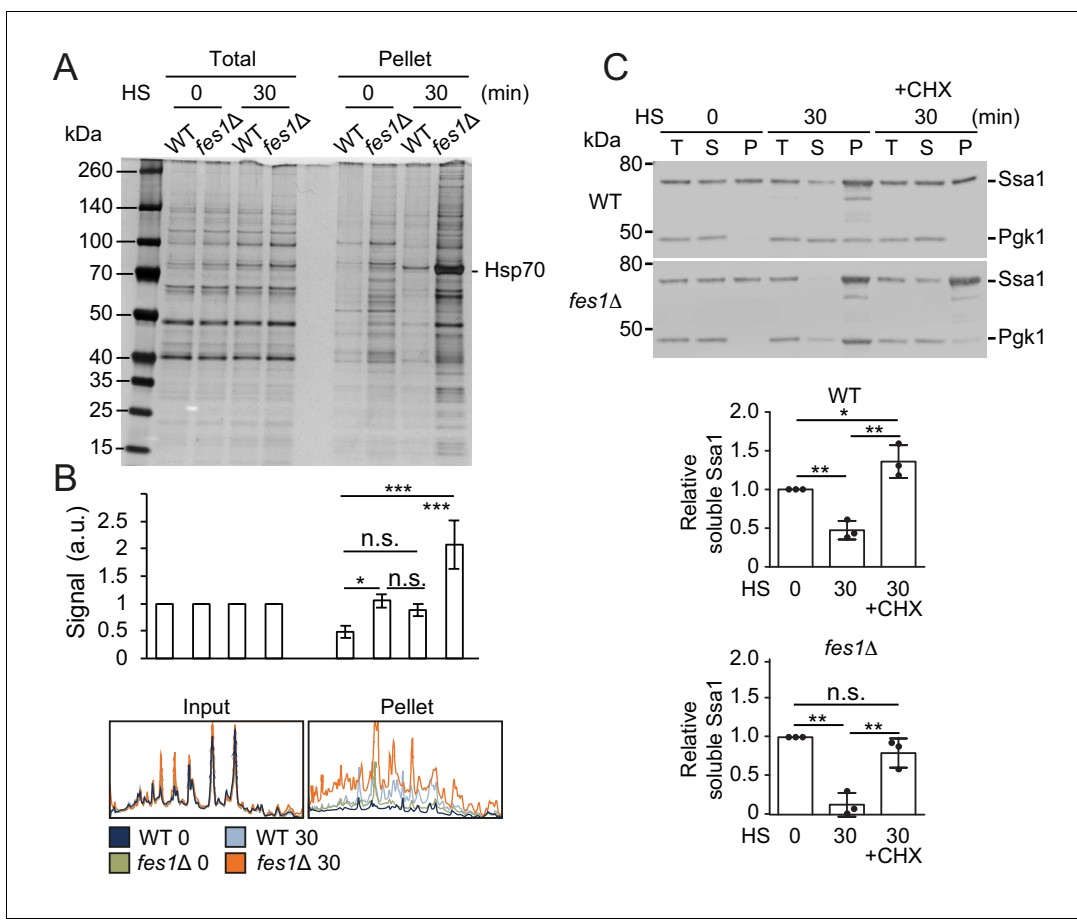

**Figure 6.** The soluble pool of Hsp70 is titrated to the aggregate fraction by heat shock. (**A**) Total lysates and the aggregate pellet fractions of protein lysates from WT and *fes1Δ* cells grown at 25°C and heat shocked for 30 min at 37°C (HS). Proteins were visualized by silver staining. (**B**) Densitometric measurements of triplicate experiments presented in A. The signals in the aggregate pellets are normalized to the respective signals in the total lysate. (**C**) Western analysis of lysates from cells as in A. Cycloheximide (CHX) was added right before the heat shock. Total lysates (**T**), the aggregate pellet (**P**) and the remaining soluble (**S**) fraction were analyzed with anti-Ssa1 and anti-Pgk1 antibodies. Relative Ssa1 signals in the soluble fractions are presented. All experiments were performed in triplicates with error bars showing standard deviation.

DOI: https://doi.org/10.7554/eLife.47791.017

The following figure supplement is available for figure 6:

**Figure supplement 1.** Heat-shock-induced protein aggregation depends on translation in WT but not *fes1Δ* cells.
DOI: https://doi.org/10.7554/eLife.47791.018

fraction upon heat-shock and that arresting translation using cycloheximide hindered both the induced aggregation and the titration of the soluble Hsp70 pool (*Figure 6C* and *Figure 6—figure supplement 1A*). On average, only 47% of the original soluble pool of Ssa1 remained after the heat shock showing efficient titration of the chaperone. When performing the analysis in *fes1Δ* cells, we found that the mutant accumulated large amounts of proteins and Hsp70 in the aggregate fraction already when cells were grown at 25°C (*Figure 6A*). Heat shocking *fes1Δ* cells resulted in extensive and widespread protein aggregation accompanied with Hsp70 almost quantitatively being transferred from the soluble to the aggregate fraction (*Figure 6A–C* and *Figure 6—figure supplement 1B*). Following the heat shock, only 13% of the original soluble pool of Ssa1 remained in the *fes1Δ* cells. Arresting translation in the *fes1Δ* cells decreased the heat-shock-induced transfer of soluble Hsp70 to the aggregate fraction but substantial aggregation still occurred. This behavior reflects the abundance of misfolded proteins that accumulate in the *fes1Δ* cells. Thus, aggregation-prone proteins sequester Hsp70 to unleash active Hsf1.

## Discussion

We have summarized our current understanding of Hsf1 regulation in *Figure 7*. Accordingly, under non-stressful conditions the lion share of constitutively trimerized Hsf1 is organized in nuclear latency complexes in which Hsp70 restricts the transcription factor from binding HSEs. Hsp70 dynamically engages the latency complexes via its canonical SBD. Under stress conditions, the pool of available Hsp70 decreases as a result of chaperone titration by the accumulation of aggregating misfolded chaperone substrate proteins. The decreased availability of Hsp70 transforms Hsf1 latency complexes into 600–700 kDa activation complexes that bind HSEs and activate transcription. The main source of titrating substrate proteins during heat shock are newly translated polypeptides. Under hyper-stress conditions, protein misfolding and aggregation become wide-spread and protein aggregates sequester Hsp70. Consequently, Hsf1 is unleashed from Hsp70 latency control. The much-increased levels of active Hsf1 promote the occupancy of HSEs, including low-affinity HSEs

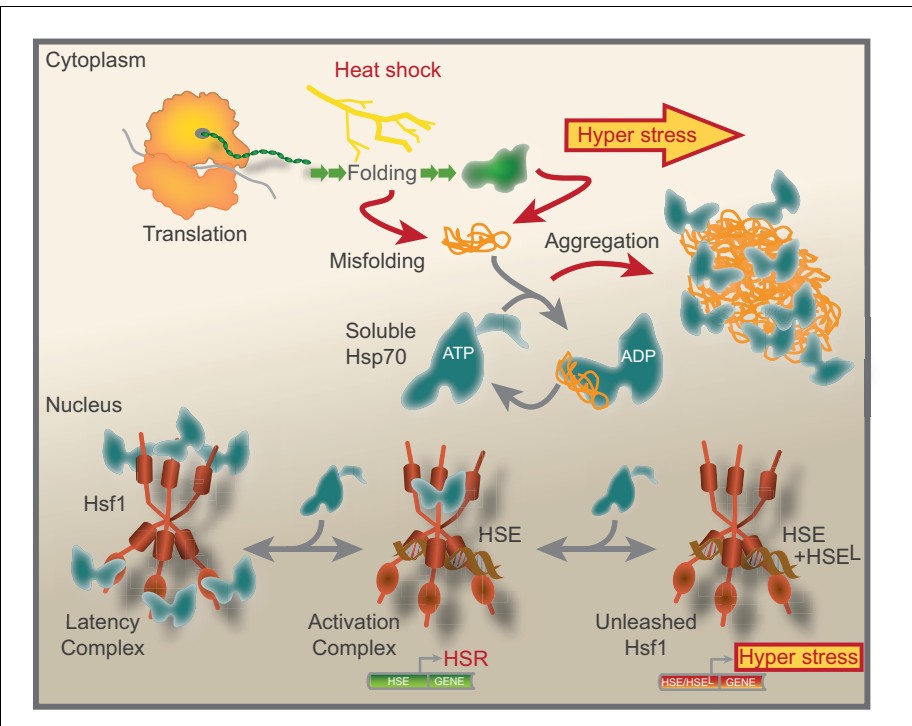

**Figure 7.** Model that show how Hsf1 is negatively regulated by the nuclear pool of Hsp70. Summative graphical model of our current understanding of Hsf1 regulation. See Discussion for details.
DOI: https://doi.org/10.7554/eLife.47791.019

(HSE$^L$), thus triggering a hyper-stress transcriptional program with a wide target signature and much increased transcriptional amplitude. Our data support a model where the nuclear pool of available Hsp70 tightly controls Hsf1 DNA-binding capacity and represses a genetic emergency program that is triggered by proteostasis collapse.

Our model is based on the nuclear localization of Hsf1 and the subcompartmentalized organization of the proteostasis system in the cytosol and nucleus as autonomous but still communicating pools. Experimentally, support of such autonomy was obtained by the finding that expression of the cytosolic NEF Sse1 in the nucleus triggers strong activation of Hsf1. Such strong deregulation of Hsf1 suggests that the nuclear and cytosolic Hsp70 pools are not immediately exchangeable via trafficking through the nuclear pores. Nevertheless, we find evidence for communication between the cytosol and nucleus. Newly translated proteins, which are produced in the cytosol are potent activators of Hsf1 when they misfold and similarly, impairing the release of persistent misfolded substrates from Hsp70 in the cytosol (fes1Δ) results in strong activation of Hsf1. Thus, the nucleus and cytosol communicate in the regulation of Hsf1, which likely involves transfer of Hsp70 and its substrates between the cytosol to the nucleus. Indeed, certain proteins that misfolded in the cytosol have been shown to target the nucleus where they are compartmentalized *en route* proteasomal degradation (*Finley et al., 2012*; *Gardner et al., 2005*; *Park et al., 2013*; *Prasad et al., 2010*; *Prasad et al., 2018*; *Samant et al., 2018*). Interestingly, the J-domain protein Sis1 is an important factor for the nuclear targeting of misfolded proteins suggesting that this Hsp70 co-chaperone is directly involved in Hsf1 regulation by delivering the titrating substrates to the nuclear Hsp70 pool (*Gardner et al., 2005*; *Prasad et al., 2010*) The implication of our data is that the regulation of Hsf1 latency relies on the compartmentalization of the proteostasis system between the cytosol and the nucleus. Nevertheless, the subcellular site(s) of Hsp70 titration by misfolded proteins remains to be determined. Upon mild heat shock we observe that half of the soluble Hsp70 (Ssa1) population is transferred to the aggregate fraction and under Hsf1 hyper-induction conditions (heat shocked *fes1Δ* cells) only 10% of the population remains soluble. Thus, the extent of the effect shows that the soluble pool of Hsp70 is titrated globally in the cell under Hsf1-inducing conditions.

Our reconstitution experiments show that Hsf1 and Hsp70 are sufficient components to form an ATP-sensitive activation complex that is competent of binding HSEs. The estimated size of the complex of 600–700 kDa and the Hsf1-Hsp70 stoichiometry of 3:1 is not readily compatible with a single trimer of Hsf1 in complex with one Hsp70 (Hsf1 = 93 kDa; Hsp70 = 70 kDa: $3 \times 93$ kDa + 70 kDa = 349 kDa). Instead the size fits better with two Hsf1 trimers each carrying a single Hsp70 ($6 \times 93$ kDa + $2 \times 70$ kDa=698 kDa). Hsf1 has previously been observed to form oligomers larger than trimers that may correspond to hexamers (*Sorger and Nelson, 1989*). Supplementing it with an excess of free Hsp70 impedes HSE binding. In light of the DNA-binding activity of the 600–700 kDa complex and our finding that dissociation of this complex by ATP does not result in increased binding of HSEs, we consider this complex to be the transcriptionally active form of Hsf1. Consistent with this notion, >669 kDa Hsf1-HSE complexes derived from yeast-cell-free lysates migrate slower on gels when mixed with Hsp70 antibodies and this effect is attenuated by ATP (*Bonner et al., 1994*). Similarly, Hsf1 complexes > 670 kDa have been observed in lysates derived from *Drosophila* and human cells (*Clos et al., 1993*; *Westwood et al., 1991*). Neither these studies nor our reconstitution address whether other proteins than Hsf1 and Hsp70 are present in the cellular activation complexes. At present, we can rule out that the J-domain protein Sis1 is a stable constituent of the complexes since we co-expressed it with Hsp70 and Hsf1 and still it did not co-purify with the activation complexes. Our in vitro data show that an excess of free Hsp70 pushes the activation complexes into latency complexes that are unable to bind HSEs. In vivo crosslinking shows that Hsp70 binds Hsf1 directly via its SBDβ and that this interaction weakens when cells are heat shocked or stressed by the buildup of misfolded proteins (AzC and *fes1Δ*). Addition of ATP to the EMSA assays did not efficiently hinder formation of the latency complexes. We do not favor the interpretation that Hsp70-ATP binds Hsf1, rather we consider that the EMSA assay relies on a gel electrophoresis step that rapidly depletes free Mg$^{2+}$ and ATP and thereby enables binding of Hsp70-ADP early during gel running. Thus, the regulatory interactions between Hsp70 and Hsf1 depend on the intrinsic Hsp70 ATPase cycle. We consider that these dynamics are central for cellular titration of the Hsp70 chaperone repressor and also explains why we obtain an activation complex rather than a latency complex when purifying Hsf1-Hsp70 from *E. coli*. Previous studies highlight the hydrophobic CE2 motif of Hsf1 as an important site for latency regulation (*Høj and Jakobsen, 1994*; *Jakobsen and*

*Pelham, 1991*; *Krakowiak et al., 2018*). Hsp70 occupancy of this and other binding sites may represent the difference between the activation and latency complexes. In support of this notion, a just published study identified an additional negative regulatory Hsp70 binding site in the N-terminal activation domain of Hsf1 (*Peffer et al., 2019*). Interestingly, the N-terminal activation domain has recently been implicated in regulating the DNA binding activity of Hsf1 (*Krakowiak et al., 2018*). Together with our data these findings suggest that multiple Hsp70-binding sites control Hsf1 affinity for DNA. Independent of the exact regulatory Hsp70-bindings sites, chaperone titration should be understood in the context of complexes between trimeric Hsf1 and multiple Hsp70s.

The chaperone repression titration model raises questions regarding what protein species functions as the stress sensor. Our finding that Hsf1 engages the canonical Hsp70 SBD indicates that the transcription factor in principle competes with any Hsp70 substrate. Newly synthesized proteins are strong candidates as the stress sensors and that upon their misfolding will function as Hsp70 titrating agents. This notion is supported by our observation that inactivation of Fes1, a cytosolic NEF that specifically releases misfolded proteins from Hsp70, triggers unsurpassed activation of Hsf1 (*Gowda et al., 2016*; *Gowda et al., 2013*; *Gowda et al., 2018*). Furthermore, blocking translation at either the initiation or elongation step diminishes Hsf1 activation in response to heat shock. In the same line, inducing misfolding of newly synthesized proteins using the proline analogue AzC activates Hsf1 and triggers synergistic potentiation when combined with heat shock. The notion that newly translated proteins are stress sensors also gains support from literature. Furthermore, it is known that newly translated proteins readily misfolds by stress. Specifically, heat shock leads to increased degradation of newly synthesized but not of long-lived cytosolic proteins and formation of aggregates decorated by Hsp104 requires ongoing translation (*Medicherla and Goldberg, 2008*; *Zhou et al., 2014*). In light of this data it is interesting to consider what kind of newly translated proteins functions as the stress sensors for Hsf1 latency regulation. Organellar and secretory proteins are prime candidates. Many are synthesized in the cytosol and are maintained in unstructured conformations by Hsp70 until they are translocated over membranes into the organelles. Consistent with the notion that these species misfold in the cytosol and regulate Hsf1, stress specific to the ER has been reported to activate Hsf1 likely as a result of hampered protein translocation (*Liu and Chang, 2008*). Similarly, mutations in the tail-anchored pathway activates Hsf1 (*Brandman et al., 2012*; *Schuldiner et al., 2008*). Another candidate class of Hsp70 titrating proteins is the ribosomal proteins and support for their involvement in Hsf1 activation has recently been obtained by experimentally impairing ribosome biogenesis (*Albert et al., 2019*; *Tye et al., 2019*). On the biochemical level, the amyloid forming peptide aβ42 that is derived from a transmembrane domain has recently been shown to decrease the interaction between Hsp70 and Hsf1 (*Zheng et al., 2016*). Nevertheless, the actual contribution of a newly translated polypeptide to Hsf1 activation will depend on how well it titrates Hsp70. This in turn is affected by a multitude of characteristics including how fast the protein folds, how chaperones impact on the folding process, how the folding is affected by the stressor, how aggregation-prone the protein is and how the misfolded species interact with Hsp70. In the end, it is likely that a plethora of different proteins function in concert to regulate Hsf1 latency via Hsp70 titration.

Our transcriptional data show that yeast cells accommodate a cryptic genetic hyper-stress program that upon activation involves a widely broadened gene-target signature and much amplified transcriptional effects. The finding shows that in WT cells the bulk of Hsf1 is maintained latent under non-stressful conditions as well as after heat shock. A simple mechanistic interpretation to explain the induction of this hyper-stress program is that the increased concentration of Hsf1 activation complexes results in intensified occupancy of HSEs, including low affinity and obstructed sites (*Erkine et al., 1999*; *Giardina and Lis, 1995*; *Santoro et al., 1998*). This interpretation explains both the broadened gene-target signature and the much-amplified transcriptional effects. Interestingly, the activation of a cryptic Hsf1 hyper-stress program in yeast is reminiscent of the induction of non-canonical and distinct genetic programs elicited by Hsf1 in cancer cells and cancer-associated fibroblasts (*Mendillo et al., 2012*; *Scherz-Shouval et al., 2014*). Specifically, the cells employ Hsf1 to induce specific oncogenic and malignancy genes that are distinct from the classical HSR target genes. Despite that the here identified hyper-stress program in yeast involves other gene targets than the mammalian programs, we speculate that the fundamental mechanism of chaperone titration may underlie the expanded Hsf1 regulons in both yeast and cancers. Accordingly, like in the yeast model also mammalian cells harbor a large pool of latent Hsf1 that can become mobilized by the

accumulation of persistent misfolded proteins that are associated with the malignant phenotype. This increased pool of constitutively active Hsf1 allows it to reach normally inaccessible genes and together with other transcription factors and chromatin modifiers to induce alternative regulons. To summarize our conceptual contribution, we find that an apparently simple chaperone-titration mechanism can produce diversified transcriptional outputs in response to distinct stress loads.

# Materials and methods

## Key resources table

| Reagent type (species) or resource | Designation | Source or reference | Identifiers | Additional information |
|---|---|---|---|---|
| Strain, strain background (*S. cerevisiae*) | AMY31 | This paper | | For details see *Table 1*. Dr. C Andréasson, Stockholm University, Sweden |
| Strain, strain background (*S. cerevisiae*) | AMY41 | This paper | | For details see *Table 1*. Dr. C Andréasson, Stockholm University, Sweden |
| Strain, strain background (*S. cerevisiae*) | AMY46 | This paper | | For details see *Table 1*. Dr. C Andréasson, Stockholm University, Sweden |
| Strain, strain background (*S. cerevisiae*) | AMY62 | This paper | | For details see *Table 1*. Dr. C Andréasson, Stockholm University, Sweden |
| Strain, strain background (*S. cerevisiae*) | BY4721 | EUROSCARF; PMID: 9483801 | Y00000 | Distributed by EUROSCARF |
| Strain, strain background (*S. cerevisiae*) | CAY1005 | EUROSCARF | Y02146 | Distributed by EUROSCARF |
| Strain, strain background (*S. cerevisiae*) | CAY1015 | PMID: 23530227 | | For details see *Table 1*. Dr. C Andréasson, Stockholm University, Sweden |
| Strain, strain background (*S. cerevisiae*) | CAY1038 | EUROSCARF | Y01514 | Distributed by EUROSCARF |
| Strain, strain background (*S. cerevisiae*) | CAY1057 | EUROSCARF | Y1512 | Distributed by EUROSCARF |
| Strain, strain background (*S. cerevisiae*) | CAY1140 | This paper | | For details see *Table 1*. Dr. C Andréasson, Stockholm University, Sweden |

*Continued on next page*

*Continued*

| Reagent type (species) or resource | Designation | Source or reference | Identifiers | Additional information |
|---|---|---|---|---|
| Strain, strain background (*S. cerevisiae*) | CAY1221 | PMID: 26912797 | | For details see *Table 1*. Dr. C Andréasson, Stockholm University, Sweden |
| Strain, strain background (*S. cerevisiae*) | CAY1255 | PMID: 14562095 | HSP104-GFP | Dr. T Nyström, University of Gothenburg, Sweden |
| Strain, strain background (*S. cerevisiae*) | CAY1257 | PMID: 23530227 | | For details see *Table 1*. Dr. C Andréasson, Stockholm University, Sweden |
| Strain, strain background (*S. cerevisiae*) | NY137 | This paper | | For details see *Table 1*. Dr. C Andréasson, Stockholm University, Sweden |
| Recombinant DNA reagent | pAM14 | This paper | | For details see *Table 2*. Dr. C Andréasson, Stockholm University, Sweden |
| Recombinant DNA reagent | pAM17 | This paper | | For details see *Table 2*. Dr. C Andréasson, Stockholm University, Sweden |
| Recombinant DNA reagent | pAM23 | This paper | | For details see *Table 2*. Dr. C Andréasson, Stockholm University, Sweden |
| Recombinant DNA reagent | pCA502 | PMID: 18948593 | | For details see *Table 2*. Dr. C Andréasson, Stockholm University, Sweden |
| Recombinant DNA reagent | pCA503 | PMID: 18948593 | | For details see *Table 2*. Dr. C Andréasson, Stockholm University, Sweden |
| Recombinant DNA reagent | pCA901 | This paper | | For details see *Table 2*. Dr. C Andréasson, Stockholm University, Sweden |

*Continued on next page*

*Continued*

| Reagent type (species) or resource | Designation | Source or reference | Identifiers | Additional information |
|---|---|---|---|---|
| Recombinant DNA reagent | pCA926 | This paper | | For details see *Table 2*. Dr. C Andréasson, Stockholm University, Sweden |
| Recombinant DNA reagent | pCA955 | PMID: 26860732 | | For details see *Table 2*. Dr. C Andréasson, Stockholm University, Sweden |
| Recombinant DNA reagent | pCA970 | PMID: 28289075 | | For details see *Table 2*. Dr. C Andréasson, Stockholm University, Sweden |
| Recombinant DNA reagent | pCA1026 | This paper | | For details see *Table 2*. Dr. C Andréasson, Stockholm University, Sweden |
| Recombinant DNA reagent | pJK001 | PMID: 28289075 | | For details see *Table 2*. Dr. C Andréasson, Stockholm University, Sweden |
| Recombinant DNA reagent | pJK010 | PMID: 28289075 | | For details see *Table 2*. Dr. C Andréasson, Stockholm University, Sweden |
| Recombinant DNA reagent | pJK011 | This work | | For details see *Table 2*. Dr. C Andréasson, Stockholm University, Sweden |
| Recombinant DNA reagent | pJK070 | PMID: 29323280 | | For details see *Table 2*. Dr. C Andréasson, Stockholm University, Sweden |
| Recombinant DNA reagent | ECYRS-BpA (plasmid) | PMID: 17560600 | | Dr. PG Schultz, The Scripps Research Institute, CA |

## Yeast strains and media

Yeast strains were derived from the BY4741 background (*Brachmann et al., 1998*). Yeast strains used in this study are listed in *Table 1*. AMY31 is a meiotic segregant obtained from a cross between CAY1195 and a BY-derivative carrying the Hsf1-EGFP-hphMX (AMY20) (*Gowda et al., 2013*). This allele was obtained by initial transformation to G418R using a PCR product amplified EGFP-kanMX amplified with primers: 5'-CCGACAGAGTACAACGATCACCGCCTGCCCAAACGAGCTAAGAAACG TACGCTGCAGGTCGAC-3' and 5'-TTAAATGATTATATACGCTATTTAATGACCTTGCCCTGTGTAC

TAATCGATGAATTCGAGCTCG-3' followed by marker swapping to hygromycinR using the hphMX cassette obtained from BamHI, SmaI, EcoRV and SpeI restricted pAG32 (*Goldstein and McCusker, 1999*). AMY41 and AMY46 are meiotic segregants obtained from a cross between BY-derivatives CAY1139 and MSN24 (*Caballero et al., 2011*; *Gowda et al., 2013*). AMY62 was obtained as a G418R clone after transformation of AMY60 with a 13 × Myc kanMX PCR product amplified from the genome of CAY1266 using primers: 5'-TTAAATGATTATATACGCTATTTAATGACCTTGCCCTG TGTACAGTATAGCGACCAGCATTC-3' and 5'- CTTATACAGTGGGCGGAGG-3' (*Gowda et al., 2013*). AMY60 is a CloNAT$^R$ derivative of CAY1211 derived from transformation using HindIII, BamHI, SacI, EcoRV restricted pAG25 (*Goldstein and McCusker, 1999*; *Gowda et al., 2018*). Strain NY137 is a HIS$^+$ derivative of AMY62 obtained by transformation with a NdeI restricted pCA926. Cells were grown in standard yeast peptone dextrose (YPD) or on ammonia-based synthetic complete dextrose (SC) medium supplemented to support the growth of auxotrophic strains.

## Plasmids

Plasmids used in this study are listed in *Table 2*. To obtain pAM14, all heat-shock responsive elements in reporter plasmid pCA955 where were mutated to STRE elements by Single Oligonucleotide Mutagenesis and Cloning Approach (SOMA) using oligonucleotide 5'- GAGAAAGTAATTAAATTA TTCCCCTTTATCTAGATCGTCCCCTCGGCGGCAAAGGGGAGAGAAAGAACCC-3' (*Masser et al., 2016*; *Pfirrmann et al., 2013*). pAM17 is a derivative of pCA955 and was constructed by yeast homologous recombination cloning according to an established protocol (*Holmberg et al., 2014*; *Masser et al., 2016*). Briefly, pCA955 was cut with SalI and KpnI and transformed with PCR products amplified from pCA955 using primers: 5'- CACCACCTTGTTCTAAAAC-3', 5'- GAGAAAGAACC-CAAAAAGAAGGTGCGCCATTTAGATTAGCC-3' and 5'- GTTGGCCGATTCATTAATGC-3', 5'- GGC TAATCTAAATGGCGCACCTTCTTTTTGGGTTCTTTCTC-3' to remove two putative stress responsive elements. To obtain pAM23 by yeast homologous recombination pRS316 was restricted with SacI, KpnI, BamHI, EcoRI and HindIII and transformed together with PCR products amplified from pCA955 (yNlucPEST) and pYM-N15 (P-*TDH3*) using primers: 5'- GTTGTAAAACGACGGCCAGTGAA TTGTAATACGACTCACTATAGGGCGCTGCTGTAACCCGTACATGC-3', 5'-CCAATCACCAACAAAA TCTTCTAAAGTAAAAACC-3' and 5'- GATTACGCCAAGCTCGGAATTAACCCTCACTAAAGGGAA-CAAAAGCTGTTAAACATTAATACGAGCAG-3', 5'-GGTTTTTACTTTAGAAGATTTGTTGGTGA TTGG-3' (*Janke et al., 2004*; *Masser et al., 2016*). Plasmid pCA901 is a derivative of pJK010 constructed by SOMA using oligonucleotides 5'-GCTGAAAAGTTGAAGAAAGTTTTGTVGACTGCTAC TAATGCCCCATTCTCTGTTGAATCC-3' and 5'-GCTGAGACAGAAGATCGTAAGTACACTCTTGCA-GAGTACATCTACACATTGCGTGG-3' (*Kaimal et al., 2017*). The integrative plasmid pCA926 was

**Table 1.** Yeast strains.

| Strain | Genotype | Reference/source |
|--------|----------|------------------|
| AMY31 | *MATa his3Δ1 leu2Δ0 ura3Δ0 HSF1-EGFP-hphMX SSA2-HA-kanMX* | This work |
| AMY41 | *MATa his3Δ1 leu2Δ0 ura3Δ0 msn2::hphMX4 msn4::natMX4 fes1Δ::LEU2* | This work |
| AMY46 | *MATa his3Δ1 leu2Δ0 ura3Δ0 msn2::hphMX4 msn4::natMX4* | This work |
| AMY62 | *MATa his3Δ1 leu2Δ0 ura3Δ0 trp1Δ::natMX Hsf1-13*Myc-kanMX* | This work |
| BY4741 | *MATa his3Δ1 leu2Δ0 met15Δ0 ura3Δ0* | (*Brachmann et al., 1998*) |
| CAY1005 | *MATa his3Δ1 leu2Δ0 met15Δ0 ura3Δ0 sse1Δ::kanMX4* | EUROSCARF |
| CAY1015 | *MATa his3Δ1 leu2Δ0 ura3Δ0* | (*Gowda et al., 2013*) |
| CAY1038 | *MATa his3Δ1 leu2Δ0 met15Δ0 ura3Δ0 hsp104Δ::kanMX4* | EUROSCARF |
| CAY1057 | *MATa his3Δ1 leu2Δ0 met15Δ0 ura3Δ0 ssa2Δ::kanMX4* | EUROSCARF |
| CAY1140 | *MATα his3Δ1 leu2Δ0 met15Δ0 ura3Δ0 fes1Δ::LEU2* | This work |
| CAY1221 | *MATa his3Δ1 leu2Δ0 ura3Δ0 fes1Δ::ura3* | (*Gowda et al., 2016*) |
| CAY1255 | *MATa his3Δ1 leu2Δ0 met15Δ0 ura3Δ0 HSP104::-GFP-his3M × 6* | This work |
| CAY1267 | *MATa his3Δ1 leu2Δ0 ura3Δ0 fes1-1 (A79R, R195A)* | (*Gowda et al., 2013*) |
| NY137 | *MATa leu2Δ0 ura3Δ0 trp1Δ::natMX Hsf1-13*Myc-kanMX his3Δ1::[SSE1-NLS HIS3]* | This work |

DOI: https://doi.org/10.7554/eLife.47791.020

**Table 2.** Plasmids.

| Plasmid | Description | Type | Reference/source |
|---|---|---|---|
| pAM14 | *URA3* P$_{CYC1-4xSTRE}$-yNlucPEST | CEN/ARS | This work |
| pAM17 | *URA3* P$_{CYC1-3xHSE}$-yNlucPEST | CEN/ARS | This work |
| pAM23 | *URA3* P$_{THD3}$-yNlucPEST | CEN/ARS | This work |
| pCA502 | *HIS3* VC | CEN/ARS | (*Andréasson et al., 2008*) |
| pCA503 | *HIS3 SSE1* | CEN/ARS | (*Andréasson et al., 2008*) |
| pCA901 | *HIS3 SSE1\*-NLS* | CEN/ARS | This work |
| pCA926 | *HIS3 SSE1-NLS* | YIP | This work |
| pCA955 | *URA3* P$_{CYC1-HSE}$-yNlucPEST | CEN/ARS | (*Masser et al., 2016*) |
| pCA970 | *HIS3 SSE1-NES* | CEN/ARS | (*Kaimal et al., 2017*) |
| pCA1026 | P$_{T7-lacO}$-6xHis-SUMO-Ssa1-S/D-Hsf1-StrepTag II P$_{T7}$-Sis1 lacI | *E. coli* | This work |
| pJK001 | *HIS3 ysfGFP-SSE1* | CEN/ARS | (*Kaimal et al., 2017*) |
| pJK010 | *HIS3 SSE1-NLS* | CEN/ARS | (*Kaimal et al., 2017*) |
| pJK011 | *HIS3 ysfGFP-Sse1-NLS* | CEN/ARS | This work |
| pJK070 | *URA3 SSA1$_{E423TAG}$-HA* | 2 µ | (*Gowda et al., 2018*) |
| ECYRS-BpA | *TRP1* BPa system | 2 µ | (*Chen et al., 2007*) |

DOI: https://doi.org/10.7554/eLife.47791.021

obtained by religating PstI restriucted pJK010. The expression plasmid pCA1026 was constructed by yeast homologous recombination involving KpnI, XhoI, BsmI and SmaI restricted pCA592, EcoRV, HindIII, KpnI and XbaI restricted pCA892 and PCR products encompassing the 1.7 kb 3' of *SSA1* (5'-GATCGATGTTGACGGTAAGC-3´ and 5'-ATTTTTTCCTCCTTTTCTCGAGTTAATCAACTTCTTC-3´), *HSF1* (5'-GAAGAAGTTGATTAACTCGAGAAAAGGAGGAAAAAATATGAATAATGCTGCAAATAC-3´ and 5'- CTATAGTGAGTCGTATTACTTCTCGAACTGCGGGTGGCTCCAGCCGCCTTTCTTAGCTCG −3´) and *SIS1-T$_{T7}$* (5'- CCACCCGCAGTTCGAGAAGTAATACGACTCACTATAGGAGTCTAAAGAGA-GAGAGAGTATGGTCAAGGAGACAAAACTT−3´ and 5'-CTAGTTATTGCTCAGCGG−3´ from template pCA892) (*Andréasson et al., 2008*; *Holmberg et al., 2014*). Appropriate tags and sequences required for bacterial expression were incorporated in the primer design. Plasmid pJK011 was obtained by SOMA of pJK001 using oligonucleotide 5'-CTGAAGGTGATGTTGACATGGACGAATTCCCGAAGAAGAAGCGGAAGGTGTAATGTTAATGCAGCAAAGTAACTAGAAAAG-3' (*Kaimal et al., 2017*). All constructed plasmids were verified by sequencing.

## Expression and purification of Hsf1-Ssa1 complexes

An overnight culture of BL21-SI/pCodonPlus transformed with pCA1026 (expresses T7-promoted 6 × His-SUMO-Ssa1, Hsf1-StrepTag II and Sis1) were diluted 100-fold in 2 × YTON supplemented with 50 mg/L kanamycin, 25 mg/L chloramphenicol and 2 mM MgSO$_4$ and grown at 30°C to OD600 0.6. 1 hr after shift to 20°C, over-night expression was induced by the addition of 0.5 mM IPTG (isopropyl β-D-thiogalactopyranoside) and 0.2 M NaCl. Cells were lysed in LWB150 (40 mM HEPES-KOH pH 7.5, 150 mM NaCl, 5 mM MgCl$_2$, 5% glycerol) supplemented with 1 mM PMSF and DNase I by two passages through an EmulsiFlex-C3 high-pressure homogenizer. The cleared supernatant (27 000 *g*; 20 min) was mixed with Protino Ni-IDA resin (Machery-Nagel GmbH and Co. KG, Düren, Germany) and the protein was allowed to bind for 5 min. After extensive washing with LWB150 the bound protein was eluted in fractions using LWB150 + 250 mM imidazole pH 7.5. The 6 × His SUMO tag was cleaved with 1% (w/w total protein) Ulp1−6 × His SUMO protease during 1 hr incubation at 20°C and the protein was bound to Strep-Tactin Sepharose (IBA Life Sciences), washed extensively and was eluted using LWB150 + 2.5 mM desthiobiotin.

## Hsf1-Ssa1 complex ATP-dissociation assay

Purified Hsf1-StrepTag II-Ssa1 complexes were incubated with different concentrations of ATP in LWB150 and was bound to Strep-Tactin Sepharose (IBA Life Sciences). After extensive washing with

LWB150, bound protein was eluted using LWB150 + 2.5 mM desthiobiotin. Samples were separated by SDS-PAGE and visualized by Pierce(TM) Silver Staining Kit (Thermo Fisher).

## Size exclusion chromatography

500 μl purified 2 mg/ml Hsf1-StrepTag II-Ssa1 complex was loaded onto a HiLoad 16/600 Superdex200 pg column (ÄKTA Fast Protein Liquid Chromatography, GE Healthcare) at a flow rate of 1 mL/min using LWB150 buffer. Eluted fractions were collected. For experiments involving supplementation with Ssa1, Hsf1-Ssa1 complexes were preincubated for 30 min at 30°C with a 10-fold molar excess of Ssa1-ATP in LWB150 buffer and loaded onto a Superose 6 10/300 GL column.

## Electrophoretic mobility shift assay

Electrophoretic mobility shift assay reactions were performed in Binding buffer (10 mM Tris-HCl pH 7.5, 50 mM KCl, 2.5 mM DTT, 0.25% Tween 20) using a final concentration of 0.05 μM annealed DY682-labeled oligo and 0.25 μM Hsf1-Ssa1 complex. The sequences of the EMSA oligos are listed in *Table 4*. The binding reactions were incubated for 20 min at 20°C in the dark before mixing with 5x Hi-Density TBE Sample buffer (450 mM Tris, 450 mM boric acid, 10 mM EDTA 0.004 mM Ficoll 400, Orange G) and running at 120V on a 6% DNA retardation gel (Invitrogen) in TBE running buffer (100 mM Tris, 100 mM boric acid, 0.2 mM EDTA) in the dark. For experiments with antibody preincubation, 5% rabbit serum was applied to a modified binding buffer (10 mM Tris-HCl pH 7.5, 50 mM KCl, 1 mM MgCl$_2$, 0.25% Tween 20) and a 0.7% agarose gel was ran in TBE buffer without EDTA. The same type of agarose gels and running conditions were also used to better visualize free HSE oligos. Gels were visualized using the Odyssey infrared imaging system (Li-COR Biosciences).

## Native PAGE

Proteins were mixed with 4 × NativePAGE Sample buffer (200 mM bis-Tris, 6 M HCl, 200 mM NaCl, 40% Glycerol, 0.004% Ponceau S, pH 7.2) and separated on NativePAGE 3–12% Bis-Tris Protein Gels (Invitrogen) using NativePAGE running buffer (50 mM bis-Tris, 50 mM Tricine pH 6.8). Gels were visualized using the Odyssey infrared imaging system (Li-COR Biosciences) before transfer to Amersham Protran Supported 0.45 μm Nitrocellulose Blotting membrane (GE Healthcare). Hsf1 signal was detected using anti-StrepTag mab, HRP conjugated (IBA Life Science) and ECL Western Blotting Analysis System (GE Healthcare) on the Odyssey infrared imaging system (Li-COR Biosciences). Protein size was determined using NativeMark unstained protein standard (Invitrogen).

## Western blot analysis and protein stability

Protein extracts were prepared from cells in logarithmic phase (*Gowda et al., 2013*; *Silve et al., 1991*). For protein stability assays, 100 mg/l cycloheximide was added before harvesting the cells. Briefly, NaOH was added directly to the cultures to a final concentration of 0.37 M and the cells were incubated for 10 min on ice before the addition of trichloroacetic acid to a final concentration of 8.3%. After centrifugation and removal of the supernatant, the pellet was rinsed with 1 M Tris base and equal amounts of SDS-solubilized protein were separated by SDS-PAGE and analyzed by quantitative western blotting, using the Odyssey Fc infrared imaging system (Li-COR Biosciences). For chemiluminescent detection (UV-crosslinking experiments) Amersham ECL Western Blotting Analysis System (GE Healthcare) and Supersignal West Dura Extended Duration Substrate (Thermo Scientific) were used. Hsf1−13 × Myc was detected with mouse anti-Myc-HRP 9E10; 1:5000 (Roche;

**Table 4.** EMSA oligos.

| Name | Sequence 5'−3' | Reference/source |
|---|---|---|
| HSE-DY682 | TCGATT**TTC**CA**GAA**CG**TTC**CATCGGC GCCGATGGAACGTTCTGGAAAATCGA | This work |
| HSE | TCGATT**TTC**CA**GAA**CG**TTC**CATCGGC GCCGATGGAACGTTCTGGAAAATCGA | This work |
| HSE* | TCGATGTGCCAGTACGTAGCATCGGC GCCGATGCTACGTACTGGCACATCGA | This work |

DOI: https://doi.org/10.7554/eLife.47791.023

RRID:AB_390910) and Ssa1$_{E423BPa}$ with rat anti-HA clone 3F10; 1:5000 (Roche; RRID:AB_390914). Nanoluc signal (anti-Nanoluc, rabbit serum; 1:1000 dilution) was normalized to the Pgk1 signal (anti-Pgk1 mouse clone 22C5D8; 1:10,000 dilution; Invitrogen).

## ChIP

ChIP experiments were carried out as previously described with minor modifications (*Cobb and van Attikum, 2010*). Briefly, 100 mL AMY31 yeast culture were grown until an OD$_{600}$ = 0.9–1 had been reached. 1% formaldehyde was added to the cultures and incubation at room temperature for 10 min, after which 125 mM glycine was added and cultures incubated for a further 5 min. Cells were harvested and washed two times with ice cold PBS. Cell breakage was done in 600 µL lysis buffer (50 mM HEPES-KOH, pH 7.5, 140 mM NaCl, 1 mM EDTA, 1% Triton X-100, 0.1% sodium deoxycholate, 1 mM PMSF, 2x Complete protease inhibitor cocktail (Roche) with zirconia beads. Cell extracts were sonicated for 15 min in the Bioruptor Plus Sonication system at high power with 30 s ON/30 s OFF cycles. Magnetic HA beads (Pierce, Cat No: 88836) were used to purify Ssa2-HA with uncoated beads as a negative control. Samples were washed 2 times with 1 mL lysis buffer, 3 times with 1 mL wash bufer (10 mM Tris-HCl, pH 8, 250 mM LiCl, 0.5% NP-40, 0.5% sodium deoxycholate, 1 mM EDTA, 1 mM PMSF), and once with 1 mL TE buffer (10 mM Tris-HCl, pH 8, 1 mM EDTA). Immuno-precipitants were recovered in 250 µL elution buffer (50 mM Tris-HCl, pH 8.0, 10 mM EDTA, 1% SDS) and treated for 2 hr at 37°C with 20 µL 10 mg/mL proteinase K (Merck). Beads were washed with 5 M LiCl, 50 mM Tris-pH 8.0 after which phenol-chloroform extraction was performed. 8 µl 5 mg/mL glycogen (Ambion) was added and after ethanol precipitation DNA, was dissolved in 30 µl ultrapure water and used for qPCR analysis using gene-specific primers. qPCR was performed using SYBR-green (KAPA) according to the manufacturer's instruction. Samples were analyzed by Rotor-Gene Q series software 1.7 using primers specific to the HSE in the *HSC82* promoter (H) and 3' end of the *HSC82* ORF (O) (See *Table 3*).

## Photo-crosslinking

AMY62, AMY63 and NY137 were co-transformed with pJK070 encoding Ssa1-HA with an amber mutation introduced at codon 423 and ECYRS-BpA for p-benzoyl-L-phenylalanine incorporation (*Chen et al., 2007*; *Gowda et al., 2018*). Transformants were grown at 30°C to mid-log phase in SC-Trp-Ura the presence of 1 mM p-benzoyl-L-phenylalanine (Bachem) added from a 100 mM stock solution freshly prepared in 1 M NaOH. Cultures were heat shocked for 3 min at 43°C or exposed to 10 mM AzC for 2 hr. Cells were rapidly harvested, washed in ice-cold water and irradiated with UV-A on ice using a Sylvania Lynx BL350 15 W fluorescent lamp for 1 hr. Total protein samples were prepared by bead beating in LWB150 or 40 mM HEPES-KOH pH 7.5, 300 mM NaCl, 1 mM EDTA, 8 M urea supplemented with 1 mM PMSF and analyzed by western blotting.

**Table 3.** qPCR primers.

| Gene | Sequence 5'→3' | Reference/source |
|------|----------------|------------------|
| *CUP1* | GTGCCAATGCCAATGTGGTAG CATTTCCCAGAGCAGCATGAC | This work |
| *HSC82 H* | CTCGTTTTCTCGAACTTC CAAATCTCCTCCCTCATTAC | This work |
| *HSC82 O* | GAGAGTTGATGAGGGTGGTG GTTAGTCAAATCTTTGACGGTC | This work |
| *TAF10* | ATATTCCAGGATCAGGTCTTCCGTAGC GTAGTCTTCTCATTCTGTTGATGTTGTTGTTG | (*Teste et al., 2009*) |
| *SSA4* | CCAAGAGGCGTACCACAAAT GCTTCTTGTTCATCTTCGGC | This work |
| *HSP104* | GTCGCTGAACCAAGTGTGAG CTCTTGCGACGGCGACACCA | This work |

DOI: https://doi.org/10.7554/eLife.47791.022

## Bioluminescent determination of Hsf1 and Msn2/4 activity

Nanoluc detection in yeast has previously been described (*Masser et al., 2016*). Briefly, Nano-Glo substrate (Promega GmbH, Germany) was diluted 1:100 with the supplied lysis buffer and mixed 1:10 with cells grown in SC in white 96-well plates. Bioluminescence was determined after 3 min incubation, using an Orion II Microplate Luminometer (Berthold Technologies GmbH and Co. KG, Germany). Bioluminescence light units (BLU) are defined as the relative light units (RLU)/s of 1 mL cells at OD600 = 1.0.

## Microscopy

Live images were taken by using a Zeiss Axiovert 200 M inverted fluorescence microscope (Carl Zeiss, Jena, Germany) with a Plan-apochromatic 63x/1.4-numerical- aperture oil immersion lens, a DG4 light source (Sutter Instruments, Novato, CA) equipped with an AxioCam MRm camera (Carl Zeiss), and SlideBook 5.0 software (Intelligent Imaging Innovations, GmbH, Göttingen, Germany). Images were acquired and processed by using SlideBook Reader software. Image quantification was done by using ImageJ software (National Institutes of Health, Bethesda, MD).

## qPCR analysis

RNA was extracted from cells grown in YPD using a RiboPure RNA Purification Kit for Yeast (Ambion, Invitrogen). cDNA was synthesized from DNase I-treated RNA using Superscript III Reverse Transcriptase (Invitrogen) and qPCR was performed using KAPA SYBR Fast Universal qPCR Kit (KAPA Biosystems) with primers listed *Table 3*. Quantification was performed using the $2^{-\Delta\Delta CT}$ method and expression was normalized to TAF10 (*Livak and Schmittgen, 2001*; *Teste et al., 2009*).

## RNA-seq analysis

Cells were grown in triplicates for each condition in YPD at 25°C until logarithmic phase and heat shocked at 37°C for 30 min in a shaking water bath. Cells were harvested by centrifugation and RNA was extracted using the RiboPure RNA Purification Kit for Yeast (Thermo Fisher Scientific). Libraries were prepared using Illumina TruSeq Stranded mRNA (polyA selection) and quality checked using Quant-iT (DNA BR) and CaliperGX. Minimum concentration was 5 ng/µl and a size of >300 base pairs. RNA-Seq was performed on Illumina HiSeq2500 (2 × 101–base pair paired end read). The sequencing reads were mapped to the sacCer2 genome with ensemble annotation *Saccharomyces cerevisiae* EF2.62 using Tophat (v 2.0.4). Gene counts and FPKMs were estimated using HTSeq (v 0.6.1) and Cufflinks (v 2.1.1), respectively. Principal component analysis was performed based on gene FPKMs using `prcomp()` function in R (v 3.5.3). Differential gene expression analysis was performed using DESeq2 (v 1.22.2). The genes with p value < 0.01, q value < 0.05 and absolute log2 fold change >1 were identified as differentially expressed.

## Isolation of protein aggregates

Protein aggregates from yeast cells were isolated by centrifugation as a detergent insoluble material (*Rand and Grant, 2006*). Cells grown to logarithmic phase was harvested by centrifugation, washed once with $H_2O$ and snap frozen in liquid nitrogen. Cells were re-suspended in ice cold Lysis buffer (100 mM Tris-HCl pH 7.5, 200 mM NaCl, 1 mM EDTA, 1 mM DTT, 5% glycerol) with 1 mM PMSF and passaged five times through an EmulsiFlex-C3 high pressure homogenizer at 25,000 psi. Cell debris was removed by 3 × 5 min centrifugation at 3000 *g* with a new tube was used for each centrifugation step. Protein concentration was adjusted to 1.5 mg/ml (Bradford assay) and 1 ml of lysate was centrifuged at 20,000 *g* for 2 × 10 min. The supernatant was removed and the pellet was twice washed with 400 µl Lysis buffer supplemented with 2 % NP-40 by centrifugation at 20,000 *g* for 10 min. The pellet was resuspended in 80 µl 4% SDS sample buffer and boiled for 4 min. 0.5% (5 µl) and 1.25% (1 µl) of the respective soluble and pelleted fractions were analyzed by SDS-PAGE and proteins were stained using the by Pierce Silver Staining Kit (Thermo Fisher).

## Data presentation and statistical testing

Data are presented as mean values of at least triplicate biologically replicated experiments (starting form independent yeast cultures) with error bars showing standard deviation. Individual data points and hence number of biological replicates are represented in the graphs as filled circles. Statistical

testing was done with Graphpad Prism seven with student t-test (pairwise comparison of distinct samples) or when one-way ANOVA multiple comparisons (for multiple comparisons). p-Values are defined as *p<0.05, **p<0.01 and ***p<0.001.

## Acknowledgements

We thank F Hurtig at the Department for help with SEC. The authors acknowledge support from Science for Life Laboratory, the Swedish National Genomics Infrastructure (NGI) and UPPMAX for providing assistance in massively parallel DNA sequencing and computational infrastructure.

## Additional information

### Funding

| Funder | Grant reference number | Author |
| --- | --- | --- |
| Swedish Cancer Society | CAN2018/711 | Claes Andréasson |
| Swedish Cancer Society | CAN2016/361 | Claes Andréasson |
| Swedish Research Council | 2015-05094 | Claes Andréasson |
| Knut och Alice Wallenbergs Stiftelse | 2017 | Claes Andréasson |
| European Research Council | Starting Grant 758397 | Marc R Friedländer |
| Swedish Research Council | 2015-04611 | Marc R Friedländer |

The funders had no role in study design, data collection and interpretation, or the decision to submit the work for publication.

### Author contributions

Anna E Masser, Conceptualization, Formal analysis, Validation, Investigation, Visualization, Methodology, Writing—original draft, Writing—review and editing; Wenjing Kang, Data curation, Formal analysis, Validation, Investigation, Methodology, Writing—review and editing; Joydeep Roy, Jany Quintana-Cordero, Validation, Investigation, Methodology, Writing—review and editing; Jayasankar Mohanakrishnan Kaimal, Conceptualization, Validation, Investigation, Methodology, Writing—review and editing; Marc R Friedländer, Data curation, Formal analysis, Supervision, Funding acquisition, Visualization, Methodology, Writing—review and editing; Claes Andréasson, Conceptualization, Resources, Supervision, Funding acquisition, Investigation, Visualization, Methodology, Writing—original draft, Project administration, Writing—review and editing

### Author ORCIDs

Claes Andréasson https://orcid.org/0000-0001-8948-0685

### Decision letter and Author response

Decision letter https://doi.org/10.7554/eLife.47791.028
Author response https://doi.org/10.7554/eLife.47791.029

## Additional files

### Supplementary files

• Transparent reporting form
DOI: https://doi.org/10.7554/eLife.47791.024

### Data availability

Sequencing data have been deposited in GEO under the previously published accession code GSE78136. This accession originally included RNA-seq data from cells grown at non-heat-shock conditions and was updated with the heat-shock data for the present study.

The following previously published dataset was used:

| Author(s) | Year | Dataset title | Dataset URL | Database and Identifier |
|---|---|---|---|---|
| Gowda NK, Kaimal JM, Masser AE, Kang W, Friedländer MR, Andréasson C | 2016 | Cytosolic splice isoform of Hsp70 nucleotide exchange factor Fes1 is required for the degradation of misfolded proteins in yeast | https://www.ncbi.nlm.nih.gov/geo/query/acc.cgi?acc=GSE78136 | NCBI Gene Expression Omnibus, GSE78136 |

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
