## [Decision Letter]

Thank you for submitting your article "Cytoplasmic protein misfolding titrates nuclear Hsp70 to activate Hsf1" for consideration by *eLife*. Your article has been reviewed by three peer reviewers, one of whom is a member of our Board of Reviewing Editors, and the evaluation has been overseen by David Ron as the Senior Editor. The following individuals involved in review of your submission have agreed to reveal their identity: Kevin Morano (Reviewer #2); David Pincus (Reviewer #3).

The reviewers have discussed the reviews with one another and the Reviewing Editor has drafted this decision to help you prepare a revised submission.

Summary:

The transcription factor Hsf1 is conserved in eukaryotes and induces expression of molecular chaperones following heat shock and other environmental stresses. Despite the deep conservation of this transcriptional circuit, the biochemical mechanisms that control Hsf1 activity have remained unclear. The Hsp70-Hsf1 regulatory circuit has been the subject of renewed interest in the last few years, and this report addresses several less well understood features of control of the ancient heat shock response (HSR) in yeast through biochemical reconstitution and in vivo analyses. The studies seek to add three important contributions to the field: (1) Hsf1 can bind Hsp70 and DNA at the same time; (2) Hsf1 is a typical Hsp70 client; (3) Excess Hsp70 removes Hsf1 from DNA; (4) Cytosolic misfolded proteins are primary Hsf1 agonists. The reviewers are in agreement, however, that additional experimental support is necessary to support these interpretations.

Essential revisions:

1) Hsf1 can bind Hsp70 and DNA at the same time:

a) A direct demonstration of Hsp70 in DNA bound complexes is essential to draw this conclusion. Specifically, Hsp70 association with Hsf1 at target gene promoters in cells by ChIP to the Ssa2 or *HSC82* promoter under non heat shock conditions and/or supershift of the DNA bound complexes assessed by EMSA in vitro would significantly strengthen this interpretation.

b) It is also not clear how the stoichiometry between Hsf1 and Hsp70 was determined to be 3:1. (The complex is 620 kDa; Hsf1 = 115 kDa; Hsp70 = 70 kDa. 3*115+70 = 415). A 3:3 stoichiometry, at 555 kDa would also be consistent with the apparent mobility. Based on the UV crosslinking, a denatured monomer of Hsf1 binds to Hsp70, so what is the evidence that a trimer only has a single Hsf1 bound?

c) The data that ATP relieves Ssa1-mediated inhibition of DNA binding is less compelling. There appears to be no effect at the 5X Ssa1 excess condition, and only minimal at 10X.

2) Hsf1 is a typical Hsp70 client:

a) The interpretation of the experiments is unclear on this point. Do the authors consider the observed effects to be due to Sse1 modulating Ssa1 nucleotide status and SBD binding to Hsf1 or to substrates in general (i.e., direct or indirect)? It is possible both scenarios are true, but they have not been distinguished. Clarification and direct experimental support, such as UV crosslinking to measure the interaction between Hsp70 and Hsf1 in WT cells treated with AzC, in Sse1-NLS cells under control conditions, and in *fes1*∆ cells under control and heat shock conditions, are needed to solidify this component of the model.

b) The subcellular site of competition for Hsp70 is unclear. The authors favor the nucleus, but no experimental support is offered. Either the model should be directly tested, or this strong interpretation should be removed from the manuscript.

c) Figure 4: the effects of AzC on both the HSR and the Msn2/4-mediated ESR have been well-established and it is not clear where the novelty lies in these experiments. Likewise the Hsp104-GFP behavior with CHX is well known – repetition/confirmation in this figure is probably unnecessary. However the leucine starvation experiments are a nice touch. Based on these data, it would be expected that Hsp104 would also not localize to puncta after leucine starvation – the authors should demonstrate this to complete the orthogonal validation of the nascent chain misfolding hypothesis.

3) Excess Hsp70 removes Hsf1 from DNA:

The data that ATP relieves Ssa1-mediated inhibition of DNA binding is less compelling. There appears to be no effect at the 5X Ssa1 excess condition, and only minimal at 10X. The lack of statistical treatment of these data (and indeed the entire manuscript) further reduces confidence in possible effects.

What is not clear is how the addition of excess Ssa1 modifies this complex to reduce DNA binding as shown in Figure 1E. What is the difference between the DNA binding-competent complexes and the so-called latency complexes? Does the molar excess of Ssa1 lead to supercomplexes with Hsf1 that could be detected by SEC or native PAGE? Does Hsf1 remain trimerized and bound to the same amount of Hsp70 after dissociating from DNA?

4) The manuscript also explore the impact of impaired cytosolic Hsp70 client release on the degree of Hsf1 de-repression. Reviewer support for this line of inquiry was significantly less enthusiastic given the fact that the authors' lab and others have previously demonstrated that cells lacking Fes1 constitutively activate the HSR specifically through Hsf1. Based on the input from reviewers, this aspect of the manuscript should be removed or alternatively extended:

a) The physiological relevance of this effect is unclear. Is there evidence that wildtype cells access this gene expression program with different levels and/or types of stress?

b) Why is this hyperactivation happening, and what are the genes that are differentially expressed and what is special about them besides the GO assignments?

c) The presentation of the RNA-seq data should be improved. The number of genes called as upregulated and downregulated seems to be an arbitrary choice of the z-score threshold, and the heat map clusters are unlabeled and therefore uninformative. A less opaque analysis would be two volcano plots showing all genes in the genome, where fold change of *fes1*∆/WT (in control and heat shock conditions) on the x-axis and a p-value for the significance based on the biological replicates on the y-axis. This will more clearly show across the genome how different the two strains are in the two conditions.

d) The authors should also examine the full sets of functionally-defined Hsf1 and Msn2/4 target genes (Pincus et al., 2018 for Hsf1, Solis et al., 2016 for Msn2/4) rather than just cherry picking 5 genes. The major result of the RNA-seq seems to be that the same genes are induced in WT and *fes1*∆, but just to a greater magnitude in the *fes1*∆ cells. This suggests that Hsf1 is just on to a higher gear – i.e., fully dissociated from Hsp70 – but not doing anything qualitatively different. A prediction here is that RNA seq in heat shocked *ssa1/2∆* double mutant would look the same as *fes1*∆. If the authors want to claim that Hsf1 has an expanded target gene repertoire in this situation, they will have to provide further evidence (e.g., ChIP Hsf1 to these new genes).

e) Figure 5J: the observation that additional Hsp70 (or at least a protein migrating near 70 kD – no antibody verification of the presumed chaperone band is provided) associates with the pellet in heat-shocked *fes1*∆ mutants is intriguing but incomplete. Namely, how much of the material observed is due to new synthesis vs. additional aggregating substrate? More importantly, is the soluble pool of Hsp70 visibly reduced? The experiment tracks input and concentrated pellet, which gives the impression that much of the Hsp70 has partitioned to the pellet. However the soluble pool composed of Ssa1/2 and the newly induced Ssa3/4 may not change.

---

## [Author Response]

Essential revisions:1) Hsf1 can bind Hsp70 and DNA at the same time:a) A direct demonstration of Hsp70 in DNA bound complexes is essential to draw this conclusion. Specifically, Hsp70 association with Hsf1 at target gene promoters in cells by ChIP to the Ssa2 or HSC82 promoter under non heat shock conditions and/or supershift of the DNA bound complexes assessed by EMSA in vitro would significantly strengthen this interpretation.

We have performed the requested experiments: Ssa2 ChIP of the *HSC82* promoter under non-heat chock conditions and the Hsf1-Ssa1 EMSA supershift experiments using anti-Ssa1 serum. The new data that lend support to the conclusion that Hsf1 can bind Hsp70 and DNA at the same time have been incorporated as Figure 1—figure supplement 1C and 1D. The manuscript has been edited to introduce the following sentences in the Results describing the new data:

“Consistent with this notion, addition of Ssa1-reactive serum resulted in that a fraction of the EMSA probe migrated as an ATP-sensitive supershifted smear (Figure 1—figure supplement 1C). In the same line, the Hsp70 Ssa2 was found to interact in vivo with the Hsf1-dependent promoter of *HSC82* by ChIP (Figure 1—figure supplement 1D).”

The Materials and methods section has been updated with information regarding the ChIP assay.

b) It is also not clear how the stoichiometry between Hsf1 and Hsp70 was determined to be 3:1. (The complex is 620 kDa; Hsf1 = 115 kDa; Hsp70 = 70 kDa. 3*115+70 = 415). A 3:3 stoichiometry, at 555 kDa would also be consistent with the apparent mobility. Based on the UV crosslinking, a denatured monomer of Hsf1 binds to Hsp70, so what is the evidence that a trimer only has a single Hsf1 bound?

To clarify, we have added new Figure 1—figure supplement 1A that presents the densitometric analysis of the Hsf1-Hsp70 complex that resulted in the stoichiometry close to 3:1. The reference to this figure has been introduced in the Results:

“Following tandem-affinity purification with matrices specific for Ssa1 (6×His-SUMO; Ni^2+^-IDA) and Hsf1 (Strep Tag II; Strep-Tactin Sepharose) a complex with the apparent stoichiometry of Hsf1 and Ssa1 3:1 was isolated (Figure 1A, Figure 1—figure supplement 1A).”

We have also clarified the part in the Discussion that describes the stoichiometry of the purified Hsf1-Hsp70 complexes (please note that yeast Hsf1 is 93 kDa):

”The estimated size of the complex of 600-700 kDa and the Hsf1-Hsp70 stoichiometry of 3:1 is not readily compatible with a single trimer of Hsf1 in complex with one Hsp70 (Hsf1 = 93 kDa; Hsp70 = 70 kDa: 3 × 93 kDa + 70 kDa = 349 kDa). Instead the size fits better with two Hsf1 trimers each carrying a single Hsp70 (6 × 93 kDa + 2 × 70 kDa = 698 kDa).”

c) The data that ATP relieves Ssa1-mediated inhibition of DNA binding is less compelling. There appears to be no effect at the 5X Ssa1 excess condition, and only minimal at 10X.

We thank the reviewers for pointing out that we may have overstated the effect of ATP addition and have now modified the description of the data in Results to better reflect the data. The experimental setup is not ideal since it relies on a preincubation with Mg^2+^-ATP but these highly charged molecules are rapidly removed from the buffer as soon as the electrical field of the gel running is applied thus potentially allowing Hsp70 to bind Hsf1.

We have modified the EMSA protocol by omitting EDTA, adding MgCl_2_ to the buffers and increasing ATP concentrations as well as used agarose instead acrylamide gels but the assays give the same result regarding HSE binding. New EMSA data has been introduced as Figure 1—figure supplement 1F. This panel shows the modest effect of ATP but importantly, the experiment now also clearly shows the increase of unbound HSE oligo upon Ssa1 addition to the Hsf1-Ssa1 complexes.

We have changed the Results to better describe the data and have included a new EMSA assays in Figure 1—figure supplement 1F:

“Interestingly, supplementation of the Hsf1-Ssa1 complexes with additional Ssa1 at low micromolar levels decreased HSE binding in a titratable manner as evidence by decreased signal of bound as well as of free oligonucleotides (Figure 1E and Figure 1—figure supplement 1E-F). With the exception of the condition in which the highest concentration of Hsp70 was applied, ATP supplementation did not significantly impact on the inhibitory effect that excess Hsp70 exerted on Hsf1 DNA binding (Figure 1E and Figure 1—figure supplement 1F).”

We have changed the Discussion to present our concerns regarding ATP supplementation in the EMSA assays:

“Addition of ATP to the EMSA assays did not efficiently hinder formation of the latency complexes. We do not favour the interpretation that Hsp70-ATP binds Hsf1, rather we consider that the EMSA assay relies on a gel electrophoresis step that rapidly depletes free Mg^2+^ and ATP and thereby enables binding of Hsp70-ADP early during gel running.”

2) Hsf1 is a typical Hsp70 client:a) The interpretation of the experiments is unclear on this point. Do the authors consider the observed effects to be due to Sse1 modulating Ssa1 nucleotide status and SBD binding to Hsf1 or to substrates in general (i.e., direct or indirect)? It is possible both scenarios are true, but they have not been distinguished. Clarification and direct experimental support, such as UV crosslinking to measure the interaction between Hsp70 and Hsf1 in WT cells treated with AzC, in Sse1-NLS cells under control conditions, and in fes1∆ cells under control and heat shock conditions, are needed to solidify this component of the model.

We have performed the Hsf1-Ssa1 UV-crosslinking experiments with AzC supplementation and in *fes1*∆ cells as well as with cells expressing Sse1-NLS. The new results have been incorporated in the manuscript as Figure 2C-F and Figure 3E. The Results section has been updated with the following sentences describing the new data and explaining or view of Sse1-NLS action:

“Western blotting demonstrated that Ssa1_E423BPa_ bound Hsf1 via its SBD under non-stressful conditions and that the interaction between Ssa1 and Hsf1 exhibited a 77% decrease when cells were heat shocked (Figure 2A-B). […] Similarly, performing the assay in *fes1*Δ cells that constitutively activate Hsf1 and accumulate misfolded proteins resulted in undetectable crosslinking (Figure 2E-F).”

On Sse1-NLS action:

“Next, we directly tested Hsp70 client binding in cells expressing Sse1-NLS by employing the Ssa1E423BPa crosslinking assay. […] These findings are consistent with the scenario that Sse1-NLS accelerates specifically the release of substrates from Hsp70 in the nucleus, including Hsf1.”

Additionally, we have included new data from protein aggregation analysis that support the notion that Sse1-NLS does not function in an indirect way by generally impairing protein folding (new Figure 3—figure supplement 1B.) The Results section has been updated to describe the new data:

“Protein aggregate analysis showed that Sse1-NLS did not trigger protein aggregation suggesting that it does not hamper general protein folding (Figure 3—figure supplement 1B).”

b) The subcellular site of competition for Hsp70 is unclear. The authors favor the nucleus, but no experimental support is offered. Either the model should be directly tested, or this strong interpretation should be removed from the manuscript.

We agree with the view that the subcellular site(s) of the Hsp70 titration presently is unclear. Our new data from aggregation preps show that a substantial fraction of the total Hsp70 associates with aggregates (see answer point 4e), suggesting large scale titration Hsp70 in both the cytosol and nucleus. Yet, Hsf1 localizes to the nucleus. Consequently, we have removed statements emphasizing that the subcellular site of Hsp70 titration by misfolded proteins is the nucleus.

The title has been changed to:

“Cytoplasmic protein misfolding titrates Hsp70 to activate nuclear Hsf1”

The sentences in the Abstract that previously mentioned the nuclear pool of Hsp70 have been edited to not include this. The sentences are now reading:

“Hsp70 binds Hsf1 via its canonical substrate binding domain and Hsp70 regulates Hsf1 DNA binding activity. During heat shock, Hsp70 is out-titrated by misfolded proteins derived from on-going translation in the cytosol.”

Similarly, the last and summarizing paragraph of the Introduction has been edited and now reads:

“During heat shock, these activating substrates are derived from the misfolding of cytosolic translation products.”

References to titration of specifically the nuclear Hsp70 pool have been removed from the Discussion (first paragraph). We have included the following new sentences:

“Nevertheless, the subcellular site(s) of Hsp70 titration by misfolded proteins remains to be determined. […] Thus, the extent of the effect shows that the soluble pool of Hsp70 is titrated globally in the cell under Hsf1 inducing conditions.”

And:

“Another candidate class of Hsp70 titrating proteins is the ribosomal proteins and support for their involvement in Hsf1 activation has recently been obtained by experimentally impairing ribosome biogenesis (Tye et al., 2019).”

The model in (now in new Figure 7) has graphically been edited not to suggest that misfolded proteins have to enter the nucleus to titrate Hsp70. To accomplish this we have removed the graphical representation of the nuclear membrane and restructured the graphics.

c) Figure 4: the effects of AzC on both the HSR and the Msn2/4-mediated ESR have been well-established and it is not clear where the novelty lies in these experiments. Likewise the Hsp104-GFP behavior with CHX is well known – repetition/confirmation in this figure is probably unnecessary.

We have moved the data that mainly confirm previous publications (old Figure 4A-C) to Figure 4—figure supplement 1. The main figure now contains data on Hsp104-GFP behavior upon CHX treatment is in combination with new data from experiments using acute leucine starvation (see below). Figure 4 has been rearranged accordingly.

However the leucine starvation experiments are a nice touch. Based on these data, it would be expected that Hsp104 would also not localize to puncta after leucine starvation – the authors should demonstrate this to complete the orthogonal validation of the nascent chain misfolding hypothesis.

We have performed the experiment of localizing Hsp104-GFP after acute leucine starvation and as predicted Hsp104 does not localize to puncta under these conditions. The new data have been introduced as Figure 4F. The Results section has been updated to describe the new data:

“Under these conditions, heat shock at 37°C did not support any detectable Hsp104-GFP foci formation (Figure 4F) and importantly, resulted in 5-fold lower levels of *SSA4* transcripts compared to the non-starved control (Figure 4G).”

3) Excess Hsp70 removes Hsf1 from DNA:The data that ATP relieves Ssa1-mediated inhibition of DNA binding is less compelling. There appears to be no effect at the 5X Ssa1 excess condition, and only minimal at 10X. The lack of statistical treatment of these data (and indeed the entire manuscript) further reduces confidence in possible effects.

We have corrected the statement and added new data, see answer for point 1c. P-values have been calculated for data throughout the manuscript including this experiment. The Materials and methods section has been updated to describe this.

What is not clear is how the addition of excess Ssa1 modifies this complex to reduce DNA binding as shown in Figure 1E. What is the difference between the DNA binding-competent complexes and the so-called latency complexes? Does the molar excess of Ssa1 lead to supercomplexes with Hsf1 that could be detected by SEC or native PAGE? Does Hsf1 remain trimerized and bound to the same amount of Hsp70 after dissociating from DNA?

We have performed new SEC analysis of the Hsf1-Ssa1 complexes and find that large supercomplexes indeed are forming upon addition of excess Ssa1. The new data have been introduced as Figure 1—figure supplement 2.

The Results have been updated with the following description of the data:

“SEC analysis of Hsf1-Ssa1 complexes revealed that addition of Ssa1 resulted in the formation of Hsf1-containing supercomplexes that eluted earlier from the column than the largest size marker (669 kDa) (Figure 1—figure supplement 2).”

The Materials and methods section has been updated with a description of the SEC experiments and figure legends have been added.

4) The manuscript also explore the impact of impaired cytosolic Hsp70 client release on the degree of Hsf1 de-repression. Reviewer support for this line of inquiry was significantly less enthusiastic given the fact that the authors' lab and others have previously demonstrated that cells lacking Fes1 constitutively activate the HSR specifically through Hsf1. Based on the input from reviewers, this aspect of the manuscript should be removed or alternatively extended:

We feel that it is important to document the global transcriptional effects of unleashing Hsf1 from Hsp70 control. Previously we and others have demonstrated that cells lacking Fes1 activate Hsf1 at normally non-stressful conditions. In contrast, the consequences for Hsf1 activity upon severe loss of soluble Hsp70 has not been investigated before. To extend and strengthen this aspect of the manuscript, we have reworked the presentation of the data (modified Figure 5 and added four new figure supplements) and added new experiments (new Figure 5J).

a) The physiological relevance of this effect is unclear. Is there evidence that wildtype cells access this gene expression program with different levels and/or types of stress?

We have extended this section by performing qPCR analysis of *CUP1* mRNA during heat shock. The promoter of *CUP1* contains a minimal low-affinity HSE that is not induced by heat-shock treatment at 37°C. Instead it is downregulated. Yet in *fes1*Δ cells it is strongly induced, making *CUP1* a diagnostic gene for the Hsf1 hyper-stress program. Using the *CUP1* marker and applying more extreme stress (45°C heat shock) we find that also wildtype cells can access the hyper-stress program.

The new data have been added as Figure 5J.

The Results section has been updated with the following text describing the finding:

“We asked if also wildtype cells could access the hyper-stress program by analysing the transcript levels of the diagnostic *CUP1* gene under more extreme heat shock conditions.[…] The *CUP1* promoter contains a minimal low affinity HSE, indicating that activation of the hyper-stress program involves increased levels of active Hsf1 due to severe out-titration of Hsp70 (Sewell et al., 1995).”

b) Why is this hyperactivation happening, and what are the genes that are differentially expressed and what is special about them besides the GO assignments?

We have clarified this section by better presentation of the GO analysis (new Figure 5—Figure supplement 4A-C).

The Results section now reads:

“GO analysis revealed that *fes1*Δ-specific induction to heat shock included genes related to cellular detoxification, response to toxic substances, oxidant detoxification and carbohydrate metabolism (Figure 5—figure supplement 4A-C). […] Overall, the behavior indicates that *fes1*Δ cells mount a hyper-stress program in response to heat-shock as a result of more induced damage by the treatment or as a result of sensitized regulatory circuits, e.g. severe Hsp70 out-titration.”

c) The presentation of the RNA-seq data should be improved. The number of genes called as upregulated and downregulated seems to be an arbitrary choice of the z-score threshold, and the heat map clusters are unlabeled and therefore uninformative. A less opaque analysis would be two volcano plots showing all genes in the genome, where fold change of fes1∆/WT (in control and heat shock conditions) on the x-axis and a p-value for the significance based on the biological replicates on the y-axis. This will more clearly show across the genome how different the two strains are in the two conditions.

We thank the reviewers for the suggestion and have revised the presentation of the RNA-seq data in Figure 5. The requested Vulcano plots are now included as Figure 5H and I. We have edited the Results section that describes the RNA-seq:

“Inspection of the expression of all genes in WT and *fes1*Δ cells before and after the heat shock, revealed a highly accentuated transcriptional response in *fes1*Δ cells involving amplified induction and inclusion of more Hsf1 and Msn2/4 target genes (Figure 5H-I and Figure 5—figure supplement 3A-B). […] The effect was even more pronounced for repressed genes with 396 identified genes in WT cells and 849 in *fes1*Δ.”

d) The authors should also examine the full sets of functionally-defined Hsf1 and Msn2/4 target genes (Pincus et al., 2018 for Hsf1, Solis et al., 2016 for Msn2/4) rather than just cherry picking 5 genes. The major result of the RNA-seq seems to be that the same genes are induced in WT and fes1∆, but just to a greater magnitude in the fes1∆ cells. This suggests that Hsf1 is just on to a higher gear – i.e., fully dissociated from Hsp70 – but not doing anything qualitatively different. A prediction here is that RNA seq in heat shocked ssa1/2∆ double mutant would look the same as fes1∆. If the authors want to claim that Hsf1 has an expanded target gene repertoire in this situation, they will have to provide further evidence (e.g., ChIP Hsf1 to these new genes).

The full set of functionally-defined Hsf1 and Msn2/4 target genes have now been included in the analysis and is presented in the new Vulcano plots in (Figure 5H-I) as well as in new density plots (Figure 5—figure supplement 3A-B). We have clarified that we indeed find that Hsf1 is on a higher gear during the hyper-stress observed in heat-shocked *fes1*Δ cells. The following key summarizing sentence has now been included in the Results section:

“Overall, the behavior indicates that *fes1*Δ cells mount a hyper-stress program in response to heat-shock as a result of more induced damage by the treatment or as a result of sensitized regulatory circuits, e.g. severe Hsp70 out-titration.”

In support of the finding that Hsf1 expands its target repertoire upon severe Hsp70 out-titration, we have also included new Figure 5J showing that Hsf1 recognizes the low-affinity HSE in the *CUP1* promoter (see answer point 4a). We have also plotted the expression all Hsf1-target genes individually under the four conditions to display how Hsf1 expands its targets when the stress is more severe (Figure 5—figure supplement 4)

e) Figure 5J: the observation that additional Hsp70 (or at least a protein migrating near 70 kD – no antibody verification of the presumed chaperone band is provided) associates with the pellet in heat-shocked fes1∆ mutants is intriguing but incomplete. Namely, how much of the material observed is due to new synthesis vs. additional aggregating substrate? More importantly, is the soluble pool of Hsp70 visibly reduced? The experiment tracks input and concentrated pellet, which gives the impression that much of the Hsp70 has partitioned to the pellet. However the soluble pool composed of Ssa1/2 and the newly induced Ssa3/4 may not change.

We have addressed the questions by performing additional aggregation experiments. The new data support to the notion that Hsp70 is titrated from the soluble to the aggregate fraction by heat shock. General aggregation as well as Hsp70 association with the aggregate fraction depend on ongoing translation in WT cells. The *fes1*Δ partially overrides this requirement, likely due to the high levels of misfolded proteins present in these cells already at the start of the heat shock.

The new data are presented in new Figure 6 and Figure 6—figure supplement 1. The Results section has been updated to present the new findings in subsection “Heat shock titrates the soluble pool of Hsp70”.

The Discussion section has been updated with:

“Upon mild heat shock we observe that half the soluble Hsp70 (Ssa1) population is transferred to the aggregate fraction and under Hsf1 hyper-induction conditions (heat shocked *fes1*Δ cells) only 10% of the population remains soluble. Thus, the extent of the effect shows that the soluble pool of Hsp70 is titrated globally in the cell under Hsf1 inducing conditions.”

Figure legends and the Materials and methods section have been updated accordingly.